# BOOSTING ADVERSARIAL TRANSFERABILITY USING DYNAMIC CUES

**Muzammal Naseer**[§*]    **Ahmad Mahmood**[○*]    **Salman Khan**[§]    **Fahad Shahbaz Khan**[§‡]

[§]Mohamed bin Zayed University of AI,    [○]Lahore University of Management Sciences
[‡]Linköping University
muzammal.naseer@alumni.anu.edu.au
(*equal contribution)

## ABSTRACT

The transferability of adversarial perturbations between image models has been extensively studied. In this case, an attack is generated from a known surrogate *e.g.*, the ImageNet trained model, and transferred to change the decision of an unknown (black-box) model trained on an image dataset. However, attacks generated from image models do not capture the dynamic nature of a moving object or a changing scene due to a lack of temporal cues within image models. This leads to reduced transferability of adversarial attacks from representation-enriched *image* models such as Supervised Vision Transformers (ViTs), Self-supervised ViTs (*e.g.*, DINO), and Vision-language models (*e.g.*, CLIP) to black-box *video* models. In this work, we induce dynamic cues within the image models without sacrificing their original performance on images. To this end, we optimize *temporal prompts* through frozen image models to capture motion dynamics. Our temporal prompts are the result of a learnable transformation that allows optimizing for temporal gradients during an adversarial attack to fool the motion dynamics. Specifically, we introduce spatial (image) and temporal (video) cues within the same source model through task-specific prompts. Attacking such prompts maximizes the adversarial transferability from image-to-video and image-to-image models using the attacks designed for image models. As an example, an iterative attack launched from image model Deit-B with temporal prompts reduces generalization (top1 % accuracy) of a video model by 35% on Kinetics-400. Our approach also improves adversarial transferability to image models by 9% on ImageNet w.r.t the current state-of-the-art approach. Our attack results indicate that the attacker does not need specialized architectures, *e.g.*, divided space-time attention, 3D convolutions, or multi-view convolution networks for different data modalities. Image models are effective surrogates to optimize an adversarial attack to fool black-box models in a changing environment over time. Code is available at https://bit.ly/3Xd9gRQ

## 1 INTRODUCTION

Deep learning models are vulnerable to imperceptible changes to the input images. It has been shown that for a successful attack, an attacker no longer needs to know the attacked target model to compromise its decisions (Naseer et al., 2019; 2020; Nakka & Salzmann, 2021). Adversarial perturbations suitably optimized from a known source model (a surrogate) can fool an unknown target model (Kurakin et al., 2016). These attacks are known as black-box attacks since the attacker is restricted to access the deployed model or compute its adversarial gradient information. Adversarial attacks are continuously evolving, revealing new blind spots of deep neural networks.

Adversarial transferability has been extensively studied in image-domain (Akhtar & Mian, 2018; Wang & He, 2021; Naseer et al., 2022b; Malik et al., 2022). Existing works demonstrate how adversarial patterns can be generalized to models with different architectures (Zhou et al., 2018) and even different data domains (Naseer et al., 2019). However, the adversarial transferability between different architecture families designed for varying data modalities, *e.g.*, image models to video models, has not been actively explored. Since the adversarial machine learning topic has gained maximum attention in the image-domain, it is natural to question if image models can help transfer

better to video-domain models. However, the image models lack dynamic temporal cues which are essential for transfer to the video models.

We are motivated by the fact that in a real-world setting, a scene is not static but mostly involves various dynamics, *e.g.*, object motion, changing viewpoints, illumination and background changes. Therefore, exploiting *dynamic cues* within an adversarial attack is essential to find blind-spots of unknown target models. For this purpose, we introduce the idea of encoding disentangled temporal representations within an image-based Vision Transformer (ViT) model using dedicated *temporal prompts* while keeping the remaining network frozen. The temporal prompts can learn the dynamic cues which are exploited during attack for improved transferability from image-domain models. Specifically, we introduce the proposed temporal prompts to three types of image models with enriched representations acquired via supervised (ViT (Dosovitskiy et al., 2020)), self-supervised (DINO (Caron et al., 2021)) or multi-modal learning (CLIP (Radford et al., 2021)).

Our approach offers the benefit that the attacks do not need to rely on specialized networks designed for videos towards better adversarial transferability. As an example, popular model designs for videos incorporate 3D convolutions, space-time attention, tube embeddings or multi-view information to be robust against the temporal changes (Bertasius et al., 2021; Arnab et al., 2021). Without access to such specific design choices, our approach demonstrates how an attacker can leverage regular image models augmented with temporal prompts to learn dynamic cues. Further, our approach can be easily extended to image datasets, where disentangled representations can be learned via tokens across a scale-space at varying image resolutions. In summary, the major contributions of this work include:

- We demonstrate how temporal prompts incorporated with frozen image-based models can help model dynamic cues which can be exploited to fool deep networks designed for videos.
- Our approach for dynamic cue modeling via prompts does not affect the original spatial representations learned by the image-based models during pre-training, *e.g.*, fully-supervised, self-supervised and multi-modal models.
- The proposed method significantly improves transfer to black-box image and video models. Our approach is easily extendable to 3D datasets via learning cross-view prompts; and image-only datasets via modeling the scale-space. Finally, it enables generalization from popular plain ViT models without considering video-specific specialized designs.

We analyse the adversarial space of three type of image models (fully-supervised, self-supervised, and text-supervised). A pre-trained ImageNet ViT with approximately 6 million parameters exhibits 44.6 and 72.2 top-1 (%) accuracy on Kinetics-400 and ImageNet validation sets using our approach, thereby significantly improving the adversarial transferability on video-domain models. A similar trend exists with other image models. Our results indicate that the multi-modal CLIP can better adapt to video modalities than fully-supervised or self-supervised ViTs. However, CLIP adversaries are relatively less transferable as compared to fully-supervised ViT or self-supervised DINO model.As an example, a momentum based iterative attack launched from our DINO model can reduce the performance of TimesFormer (Bertasius et al., 2021) from 75.6% to 35.8% on Kinetics-400 dataset.

## 2 BOOSTING ADVERSARIAL TRANSFERABILITY USING DYNAMIC CUES

Adversarial transferability refers to manipulating a clean sample (image, video, or 3D object rendered into multi-views) in a way that is deceiving for an unknown (black-box) model. In the absence of an adversarial perturbation, the same black-box model predicts the correct label for the given image, video, or a rendered view of a 3D object. A known surrogate model is usually used to optimize for the adversarial patterns. Instead of training the surrogate model from scratch on a given data distribution, an attacker can also adapt pre-trained image models to the new task. These image models can include supervised ImageNet models such as Deit (Touvron et al., 2020), self-supervised ImageNet models like DINO (Caron et al., 2021), and text-supervised large-scale multi-modal models *e.g.* CLIP (Radford et al., 2021). The adversarial attack generated from such pre-trained models with enriched representations transfer better in the black-box setting for image-to-image transfer task (Zhang et al., 2022; Naseer et al., 2022b; Aich et al., 2022). However, adversarial perturbations optimized from image models are not well suited to fool motion dynamics learned by a video model (Sec. 3). To cater for this, we introduce temporal cues to model motion dynamics within adversarial attacks through pre-trained image models. Our approach, therefore, models both spatial and temporal

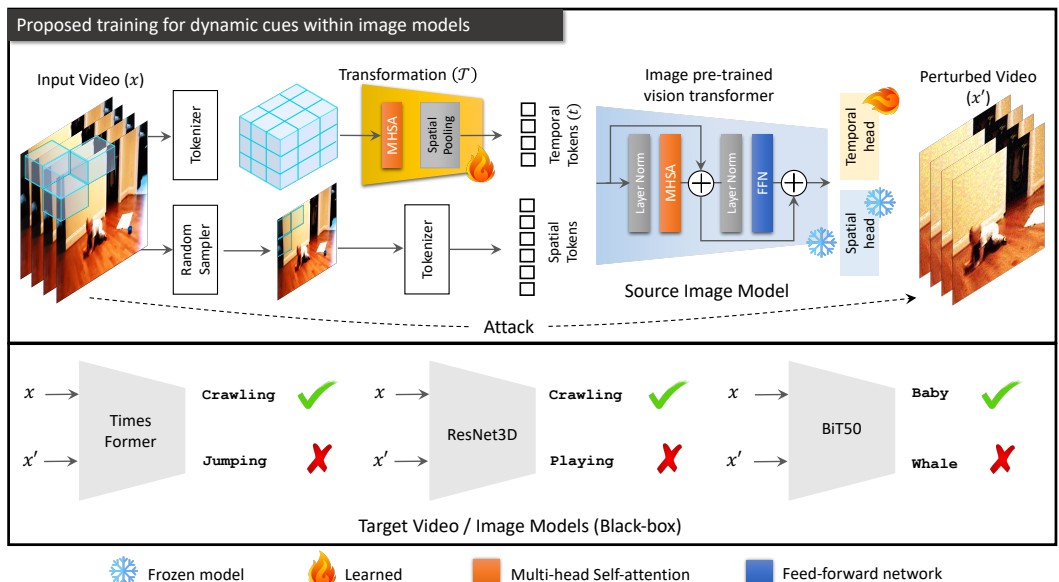

Figure 1: **Overview of inducing dynamic cues within image models:** Attackers can easily access freely available, pre-trained image models learned on large-scale image and language datasets to launch adversarial attacks. These models, however, lack temporal information. Therefore, adversarial attacks launched using image models have less success rate against a *moving target* such as in videos. We learn a transformation $\mathcal{T}(.)$ to convert a given video with $t$ number of frames into $t$ temporal tokens. Our transformation is based on self-attention thus it learns the motion dynamics between the video frames with global context during training. We randomly sample a single frame to represent the spatial tokens. The temporal and spatial tokens are concatenated and passed through the frozen model. The spatial tokens are ignored while the average of temporal tokens is processed through a temporal head for video recognition. The image/spatial class token learned on images (*e.g.*, ImageNet) interacts with our temporal tokens (*e.g.*, Kinetics) within the network through self-attention. After the training, image and video solutions are preserved within the spatial and temporal tokens and used in adversarial attacks.

information during the attack (Sec. 2.1) and does not depend on specialized video networks *e.g.*, with 3D convolutions (Tran et al., 2018), space-time divided attention (Bertasius et al., 2021) or even multi-branch models (Su et al., 2015). To this end, we first introduce *temporal prompt adaptation* of image models to learn dynamic cues on videos or multi-view information on rendered views of a 3D object, as explained next.

## 2.1 Introducing Dynamic Cues through Temporal Prompts

**Preliminaries:** We consider an image model $\mathcal{F}$ pre-trained on the input samples $x$ of size $c \times h \times w$, where $c, h$ and $w$ represent the color channels, height and width, respectively. We consider Vision Transformers (ViTs) (Dosovitskiy et al., 2020; Touvron et al., 2020) sequentially composed of $n$ Transformer blocks comprising of multi-head self-attention (MHSA) and feed-forward layer (Dosovitskiy et al., 2020) *i.e.* $\mathcal{F} = (f_1 \circ f_2 \circ f_3 \circ \ldots f_n)$, where $f_i$ represents a single Transformer block. ViTs divide an input sample $x$ into $N$ patches also called patch tokens, $P_t \in \mathbb{R}^{N \times D}$, where $D$ is the patch dimension. A class token[1], $\mathcal{I}_{cls} \in \mathbb{R}^{1 \times D}$, is usually combined with these patch tokens within these image models (Dosovitskiy et al., 2020; Caron et al., 2021; Radford et al., 2021). We refer to these patch and class tokens collectively as '*spatial tokens*' (Fig. 1). These tokens are then processed by the model $\mathcal{F}$ to produce an output of the same size as of input *i.e.* $\mathcal{F}(x) \in \mathbb{R}^{(N+1) \times D}$. In the case of image classification, the refined class token ($\tilde{\mathcal{I}}_{cls}$) is extracted from the model output and processed by MLP ($g_s$), named spatial head (Fig. 1). It maps the refined class token from $\mathbb{R}^{1 \times D}$ to $\mathbb{R}^{1 \times C_s}$, where $C_s$ represents the number of image class categories.

**Motivation:** The spatial class token in image models, $\mathcal{I}_{cls}$, is never optimized for temporal information. Our objective is to adapt the pre-trained and frozen image model for videos or multi-views by training a temporal class token $\mathcal{I}_{cls}^t \in \mathbb{R}^{1 \times D}$ that is optimal for input samples of size $t \times c \times h \times w$,

---

[1]Hierarchical ViTs such as Swin Transformer (Liu et al., 2021c) or more simple designs without self-attention layers like MLP-Mixer (Tolstikhin et al., 2021) use average of patch tokens as the class token.

Figure 2: *Mimicking dynamic cues on static images:* We optimize our proposed transformation (Fig. 1) for different spatial scales to mimic changes. This improves model generalization *e.g.* Deit-T performance increases from 7.7% to 45% on ImageNet val. set at 96×96. Attacking models with such prompts increases transferability.

where $t$ represents the number of temporal frames in a video or rendered views of a 3D object. We optimize the temporal token through MLP ($g_t$), named temporal head (Fig. 1). It maps the temporal token from $\mathbb{R}^{1 \times D}$ to $\mathbb{R}^{1 \times C_t}$, where $C_t$ represents the number of video action categories.

A trivial way is to concatenate all the additional frames of a video to generate $t \times N$ patch tokens which can then be combined with the temporal token before forwadpass through the image model. We can then optimize temporal class token $\mathcal{I}_{cls}^t$ and temporal head ($g_t$) for video classification and incorporate the dynamic cues within a frozen image model. This approach, however, has a drawback as the computational time complexity increases significantly to $\mathcal{O}\left((t \times N)^2 \times D\right)$ within the self-attention layers of ViTs. Another naive way is to either apply temporal pooling to convert $t \times N$ to only $N$ spatial tokens or approximate a video via a single frame that is $t_i \times c \times h \times w$, where $t_i$ corresponds to a randomly sampled frame from the input video. These approaches are however sub-optimal as we are unable to optimize for motion dynamics available from different video frames.

We induce the temporal information within image models without increasing the quadratic complexity within self-attention of a Vision Transformer. We achieve this by representing a given video by a randomly sampled frame while at the same time each frame in a video is modeled via a temporal prompt of size $\mathbb{R}^{1 \times D}$. The temporal prompts for different frames in a video are generated through our transformation $\mathcal{T}$. Therefore, we only train transformation $\mathcal{T}$, temporal class token $\mathcal{I}_{cls}^t$ and head $g_t$ to learn motion dynamics within pre-trained and frozen image models. Our approach preserves the original image or spatial representation encoded within spatial class token $\mathcal{I}_{cls}$ (Table 5). Both spatial and temporal tokens, $\mathcal{I}_{cls}$ and $\mathcal{I}_{cls}^t$, are then used within our attack approach (Sec. 2.2).

### 2.1.1 TEMPORAL PROMPTS THROUGH TRANSFORMATION

As mentioned above, the transformation $\mathcal{T}$ processes a video with $t$ number of temporal frames and produces only $t$ temporal prompts (Fig. 1). A video sample, $\boldsymbol{x}$, is divided into patch tokens such that $\boldsymbol{x} \in \mathbb{R}^{t \times N \times D}$. The transformation $\mathcal{T}$ is a single self-attention block (Dosovitskiy et al., 2020) that interacts between the tokens and learns the relationship between the video frames. The transformation output is then pooled across spatial dimension $N$ to produce $t$ temporal tokens *i.e.*, $\mathcal{T}(\boldsymbol{x}) \in \mathbb{R}^{t \times D}$. Attacking such temporal tokens allows finding adversarial patterns capable of fooling the dynamic cues learned by the unknown black-box video model. **Mimicking dynamic behavior on image datasets:** Image samples are static and have no temporal dimension. So we adopt a simple strategy to mimic changing behavior for such static data. We consider images at different spatial scales to obtain a scale-space (Fig. 2) and learn prompts for different spatial scales. The dynamic cues within image models not only boost transferability of the existing adversarial attacks from image-to-video models but also increase the attack success rate in the black-box image models (Sec. 3).

### 2.1.2 IMAGE MODELS WITH TEMPORAL AND SPATIAL PROMPTS

Our approach is motivated by the shallow prompt transfer learning (Jia et al., 2022). As discussed above, we divide a given a video $\boldsymbol{x}$ into video patches such that $\boldsymbol{x} \in \mathbb{R}^{t \times N \times D}$. These video patches are processed by our transformation to produce temporal tokens *i.e.* $\mathcal{T}(\boldsymbol{x}) \in \mathbb{R}^{t \times D}$. Further, we randomly sample a single frame $\boldsymbol{x}_i \sim \boldsymbol{x}$ and divide it into patch tokens such that $\boldsymbol{x}_i \in \mathbb{R}^{N \times D}$. These single frame tokens are then concatenated with the temporal prompts generated by $\mathcal{T}(\boldsymbol{x})$, a temporal class token $\mathcal{I}_{cls}^t$, and the image class token $\mathcal{I}_{cls}$ to equip the pre-trained image representation of a model $\mathcal{F}$ with discriminative temporal information.

$$\mathcal{F}(\boldsymbol{x}) = \mathcal{F}\left(\left[\mathcal{I}_{cls}^t, \mathcal{T}(\boldsymbol{x}), \mathcal{I}_{cls}, \boldsymbol{x}_i\right]\right) \tag{1}$$

After the forwardpass through the image model (Eq. 1), we extract the refined temporal class token and temporal prompts from the model's output. The average of these refined temporal class tokens,

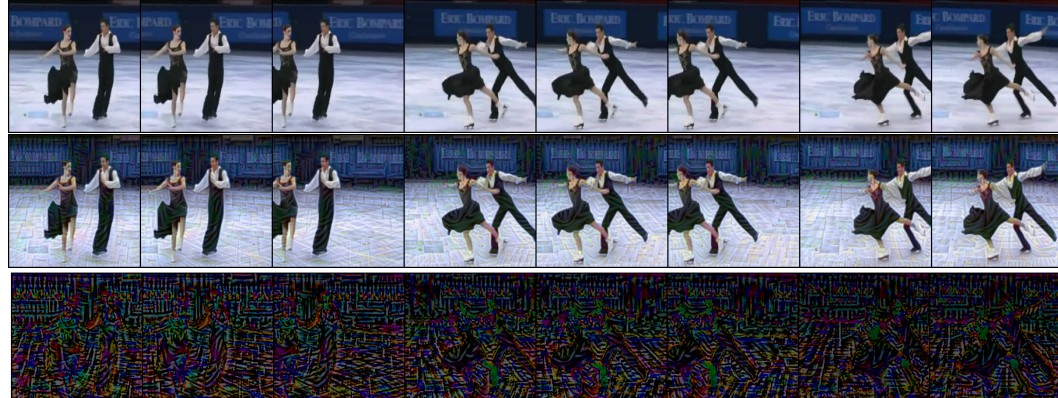

Figure 3: *Visualizing the behavior of Adversarial Patterns across Time:* We generate adversarial signals from our DINO model with temporal prompts using DIM attack (Xie et al., 2019). Observe the change in gradients across different frames (*best viewed in zoom*). We provide visual demos of adversarial patterns in Appendix K.

and $t$ temporal prompts is then projected through the temporal head $g_t$ for the new video classification task. For the sake of brevity, we will refer to the average of our refined temporal class and prompts tokens collectively as $\tilde{\mathcal{I}}_{cls}^{tp}$. During backpass, we only update the parameters of our transformation, temporal class token and temporal head (Fig. 1). **Training:** Given a pre-trained ViT, we freeze all of its existing weights and insert our learnable transformation ($\mathcal{T}$), temporal class token $\mathcal{I}_{cls}^{t}$, and video-specific temporal head. We train for 15 epochs only using SGD optimizer with a learning rate of $0.005$ which is decayed by a factor of 10 after the $11^{\text{th}}$ and $14^{\text{th}}$ epoch. We use batch size of 64 and train on 16 A100 GPUs for large-scale datasets such as Kinetics-400 (Kay et al., 2017) and only 2 A100 GPUs for other small datasets. We discuss the effect of our temporal prompts on temporal (video) and spatial (image) solutions in Table 5. Our method mostly retains the original image solution captured in image class token $\mathcal{I}_{cls}$ while exhibiting dynamic temporal modeling.

## 2.2 TRANSFERABLE ATTACK USING SPATIAL AND TEMPORAL CUES

Given an image model $\mathcal{F}$ adapted for motion dynamics via our approach (Fig. 1), our attack exploits the spatial and temporal cues to generate transferable perturbations for both image and video models.

- *Given a video and its corresponding label:* Our attack uses the refined temporal class token $\tilde{\mathcal{I}}_{cls}^{tp}$ output by the model in a supervised manner, while the spatial class token $\tilde{\mathcal{I}}_{cls}$ serves in an self-supervised objective.

- *Given an image and its corresponding label:* Our attack uses the refined class tokens $\tilde{\mathcal{I}}_{cls}^{tp}$ learned at different resolutions to mimic the dynamic behavior from images in a supervised manner, while the spatial class token $\tilde{\mathcal{I}}_{cls}$ serves in an self-supervised objective.

We generate an adversarial example $\boldsymbol{x'}$ for a video or image sample $\boldsymbol{x}$ using the following objective:

$$\text{find} \quad \boldsymbol{x'} \quad \text{such that} \quad \mathcal{F}(\boldsymbol{x'})_{argmax} \neq y, \quad \text{and} \quad \|\boldsymbol{x} - \boldsymbol{x'}\|_p \leq \epsilon, \tag{2}$$

where $y$ is the original label and $\epsilon$ represents a maximum perturbation budget within a norm distance $p$. We optimize Eq. 2 by maximizing the loss objective in Eq. 3 within existing adversarial attacks.

$$\text{maximize} \quad \mathcal{L} = \mathcal{L}_s - \mathcal{L}_{ss}, \tag{3}$$

where $\mathcal{L}_s$ is a supervised loss function. It uses the labels of a given image/video through supervised MLP head. $\mathcal{L}_s$ can be optimized by any of the existing supervised losses proposed in transferable attacks literature; cross-entropy (Dong et al., 2018), relativistic cross-entropy (Naseer et al., 2019), or logit loss (Zhao et al., 2021). Unless otherwise mentioned, we use cross-entropy loss to demonstrate the effectiveness of our approach. Similarly, $\mathcal{L}_{ss}$ is self-supervised loss that exploits pre-trained representation within spatial class tokens. We minimize the cosine similarity between the refined class tokens of clean and adversarial samples. Specifically, we define $\mathcal{L}_s$ and $\mathcal{L}_{ss}$ as follows:

$$\mathcal{L}_s = -\sum_{j=1}^{C_t} y_j log \left( \tilde{\mathcal{I}'}_{cls}^{tp} \circ g_t \right), \quad \mathcal{L}_{ss} = \frac{\tilde{\mathcal{I}'}_{cls}^{\top} \tilde{\mathcal{I}}_{cls}}{\|\tilde{\mathcal{I}'}_{cls}\|\|\tilde{\mathcal{I}}_{cls}\|}$$

| | | **Fast Gradient Sign Method (FGSM) (Goodfellow et al., 2014)** | | | | | | | | | |
|---|---|---|---|---|---|---|---|---|---|---|---|
| | | TimesFormer | | | | ResNet3D | | | | MVCNN | |
| Models (↓) | Temporal Prompts | UCF | HMDB | K400 | SSv2 | UCF | HMDB | K400 | SSv2 | Depth | Shaded |
| | | 90.6 | 59.1 | 75.6 | 45.1 | 79.9 | 45.4 | 60.5 | 26.7 | 92.6 | 94.8 |
| Deit-T | ✗ | 75.1 | 32.9 | 57.8 | 23.5 | 25.7 | 9.8 | 23.3 | 14.7 | 16.0 | 79.0 |
| | ✓ | 64.5$_{(-26.1)}$ | 21.9$_{(-37.2)}$ | 51.6$_{(-24.0)}$ | 19.8$_{(-25.3)}$ | 22.3$_{(-57.6)}$ | 8.9$_{(-36.5)}$ | 19.4$_{(-41.1)}$ | 13.9$_{(-12.8)}$ | 15.2$_{(-77.4)}$ | 77.4$_{(-17.4)}$ |
| Deit-S | ✗ | 74.7 | 33.5 | 58.2 | 23.0 | 25.4 | 9.7 | 23.9 | 15.9 | 17.2 | 79.8 |
| | ✓ | 64.0$_{(-26.6)}$ | 20.6$_{(-38.5)}$ | 48.5$_{(-27.1)}$ | 19.1$_{(-26.0)}$ | 22.6$_{(-57.3)}$ | 8.6$_{(-36.8)}$ | 20.6$_{(-39.9)}$ | 13.6$_{(-13.1)}$ | 15.9$_{(-76.7)}$ | 78.9$_{(-15.9)}$ |
| Deit-B | ✗ | 75.5 | 31.7 | 57.5 | 22.6 | 25.0 | 9.2 | 22.4 | 15.5 | 17.3 | 80.6 |
| | ✓ | 64.7$_{(-25.9)}$ | 20.0$_{(-39.1)}$ | 48.6$_{(-27.0)}$ | 19.3$_{(-25.8)}$ | 22.7$_{(-57.2)}$ | 8.1$_{(-37.3)}$ | 19.5$_{(-41.0)}$ | 13.4$_{(-13.3)}$ | 17.1$_{(-75.5)}$ | 80.9$_{(-13.9)}$ |
| DINO | ✗ | 71.0 | 33.7 | 54.2 | 23.8 | 32.0 | 12.4 | 24.3 | 15.2 | 15.2 | 72.7 |
| | ✓ | 60.7$_{(-29.9)}$ | 18.8$_{(-40.3)}$ | 46.7$_{(-28.9)}$ | 19.4$_{(-25.7)}$ | 21.9$_{(-58.0)}$ | 8.3$_{(-37.1)}$ | 19.6$_{(-40.9)}$ | 13.9$_{(-12.8)}$ | 15.0$_{(-77.6)}$ | 70.7$_{(-24.1)}$ |
| CLIP | ✗ | 79.8 | 36.6 | 62.3 | 26.4 | 38.5 | 15.0 | 29.9 | 17.3 | 15.8 | 81.4 |
| | ✓ | 77.5$_{(-13.1)}$ | 29.3$_{(-29.8)}$ | 59.5$_{(-16.1)}$ | 25.0$_{(-20.1)}$ | 35.6$_{(-44.3)}$ | 11.2$_{(-34.2)}$ | 28.8$_{(-31.7)}$ | 16.5$_{(-10.2)}$ | 15.6$_{(-77.0)}$ | 81.2$_{(-13.6)}$ |
| | | **Projected Gradient Decent (PGD) (Madry et al., 2018)** | | | | | | | | | |
| Deit-T | ✗ | 84.9 | 40.9 | 68.3 | 32.5 | 69.4 | 32.7 | 44.3 | 18.4 | 37.0 | 90.5 |
| | ✓ | 78.1$_{(-12.5)}$ | 32.0$_{(-27.1)}$ | 61.5$_{(-14.1)}$ | 27.8$_{(-17.3)}$ | 65.9$_{(-14.0)}$ | 29.5$_{(-15.9)}$ | 39.1$_{(-21.4)}$ | 17.6$_{(-9.1)}$ | 34.6$_{(-58.0)}$ | 90.2$_{(-4.6)}$ |
| Deit-S | ✗ | 85.1 | 42.0 | 68.6 | 32.5 | 69.9 | 32.7 | 44.8 | 19.4 | 39.4 | 90.9 |
| | ✓ | 74.1$_{(-16.5)}$ | 26.1$_{(-33.0)}$ | 58.3$_{(-17.3)}$ | 26.1$_{(-19.0)}$ | 65.4$_{(-14.5)}$ | 28.2$_{(-17.2)}$ | 28.9$_{(-31.6)}$ | 17.1$_{(-9.6)}$ | 36.5$_{(-56.1)}$ | 90.1$_{(-4.7)}$ |
| Deit-B | ✗ | 85.5 | 41.5 | 67.7 | 33.3 | 69.9 | 32.6 | 44.7 | 18.9 | 37.7 | 91.1 |
| | ✓ | 73.6$_{(-17.0)}$ | 25.0$_{(-34.1)}$ | 56.3$_{(-19.3)}$ | 24.4$_{(-20.7)}$ | 64.7$_{(-15.2)}$ | 26.6$_{(-18.8)}$ | 37.0$_{(-23.5)}$ | 16.6$_{(-10.1)}$ | 36.7$_{(-55.9)}$ | 90.5$_{(-4.3)}$ |
| DINO | ✗ | 81.7 | 38.0 | 64.8 | 30.9 | 68.0 | 31.8 | 42.5 | 18.2 | 33.1 | 89.2 |
| | ✓ | 64.9$_{(-25.7)}$ | 14.8$_{(-44.3)}$ | 51.5$_{(-24.1)}$ | 22.4$_{(-22.7)}$ | 63.1$_{(-16.8)}$ | 25.9$_{(-19.5)}$ | 35.2$_{(-25.3)}$ | 16.6$_{(-10.1)}$ | 28.9$_{(-63.5)}$ | 87.5$_{(-7.3)}$ |
| CLIP | ✗ | 86.9 | 44.5 | 70.7 | 35.4 | 71.9 | 35.1 | 46.5 | 20.0 | 39.8 | 91.3 |
| | ✓ | 82.6$_{(-8.0)}$ | 32.3$_{(-26.8)}$ | 66.9$_{(-8.7)}$ | 32.7$_{(-12.4)}$ | 69.7$_{(-10.2)}$ | 28.9$_{(-16.5)}$ | 43.2$_{(-17.3)}$ | 18.9$_{(-7.8)}$ | 37.6$_{(-55.0)}$ | 90.1$_{(-4.7)}$ |

Table 1: *Adversarial Transferability from image to video models:* Adversarial accuracy (%) is reported at $\epsilon \leq 16$. When attack is optimized from an image model without temporal prompts (✗), then we minimize the unsupervised adversarial objective (Eq. 3) i.e., cosine similarity between the feature embedding of spatial class token of clean and adversarial frames. We use 8 frames from a given video. Attacks transferability improves by a clear margin with temporal prompts. Clean accuracy (top1 % on randomly selected validation set of 1.5k samples) is highlighted with cell color. *Single step FGSM and multi-step PGD attacks perform better with our proposed temporal prompts (e.g. FGSM reduces TimesFormer generalization from 90.6% to 64.5% on UCF by exploiting dynamic cues within the DeiT-T). A similar trend exists for ResNet3D and MVCNN.*

In this manner, any of the existing attacks can be combined with our approach to transfer adversarial perturbations as we demonstrate in Sec. 3.

### 2.2.1 CHANGING SPATIAL RESOLUTION FOR REGULARIZATION

An adversarial attack can easily overfit the source models meaning it can have a 100% success rate on the source model but mostly fails to fool the unknown black-box model. Different heuristics like adding momentum (Dong et al., 2018; Lin et al., 2019; Wang & He, 2021), augmentations (Naseer et al., 2021; Xie et al., 2019), ensemble of different models (Dong et al., 2018) or even self-ensemble (Naseer et al., 2022b) at a target resolution are proposed to reduce such overfitting and increase adversarial transferability. Low-resolution pre-training followed by high-resolution fine-tuning have a regularization effect on the generalization of neural networks (Touvron et al., 2022). Similarly, we observe that fine-tuning scale-space prompts at different low resolutions (e.g., 96×96) within a model pre-trained on high resolution (224×224) also has a complimentary regularization effect on the transferability of adversarial perturbations (Tables 3 and 4). Most publicly available image models are trained at a resolution of size 224×224 and their performance at low-resolution inputs is sub-optimal (Fig. 2). Learning at low-resolution with our approach (Fig. 1) also allows to mimic a changing scene over time from static datasets like ImageNet (Fig. 2). Our low-resolution scale-space prompts (Sec. 2.1) significantly improves generalization at low-resolution inputs (Fig. 2).

## 3 EXPERIMENTS

We generate $l_\infty$ adversarial examples from the adapted image models and study their transferability to video and image models. We followed standard adversarial transferability protocol Dong et al. (2018); Naseer et al. (2022b); the attacker is unaware of the black-box video model but has access to its train data. Our temporal cues also boost cross-task adversarial transferability with no access

| Models (↓) | Temporal Prompts | TimesFormer | | | | ResNet3D | | | | MVCNN | |
|---|---|---|---|---|---|---|---|---|---|---|---|
| | | UCF | HMDB | K400 | SSv2 | UCF | HMDB | K400 | SSv2 | Depth | Shaded |
| | | 90.6 | 59.1 | 75.6 | 45.1 | 79.9 | 45.4 | 60.5 | 26.7 | 92.6 | 94.8 |
| Deit-T | ✗ | 77.0 | 30.1 | 58.5 | 25.2 | 42.0 | 16.2 | 29.1 | 16.0 | 19.2 | 83.0 |
| | ✓ | 66.9$_{(-23.7)}$ | 21.4$_{(-37.7)}$ | 51.4$_{(-24.2)}$ | 21.5$_{(-23.6)}$ | 41.5$_{(-38.4)}$ | 16.1$_{(-29.3)}$ | 26.4$_{(-34.1)}$ | 15.3$_{(-11.4)}$ | 19.0$_{(-73.6)}$ | 82.9$_{(-11.9)}$ |
| Deit-S | ✗ | 78.7 | 31.8 | 59.3 | 25.1 | 42.2 | 16.5 | 31.0 | 16.5 | 20.6 | 83.6 |
| | ✓ | 60.8$_{(-29.8)}$ | 16.1$_{(-43.0)}$ | 47.6$_{(-28.0)}$ | 21.0$_{(-24.1)}$ | 41.9$_{(-38.0)}$ | 15.7$_{(-29.7)}$ | 26.8$_{(-33.7)}$ | 14.2$_{(-12.5)}$ | 19.3$_{(-73.3)}$ | 83.5$_{(-11.3)}$ |
| Deit-B | ✗ | 78.7 | 32.0 | 59.7 | 25.9 | 43.1 | 16.2 | 29.5 | 16.8 | 20.6 | 84.4 |
| | ✓ | 61.7$_{(-25.9)}$ | 15.6$_{(-43.5)}$ | 45.2$_{(-30.4)}$ | 19.1$_{(-26.0)}$ | 40.7$_{(-39.2)}$ | 13.6$_{(-31.8)}$ | 24.8$_{(-35.7)}$ | 14.5$_{(-12.2)}$ | 20.5$_{(-72.1)}$ | 84.2$_{(-10.6)}$ |
| DINO | ✗ | 71.3 | 28.0 | 54.6 | 24.1 | 46.0 | 18.8 | 30.3 | 15.9 | 19.7 | 80.4 |
| | ✓ | 47.7$_{(-42.9)}$ | 8.40$_{(-50.7)}$ | 38.8$_{(-36.8)}$ | 16.4$_{(-28.7)}$ | 39.8$_{(-40.1)}$ | 12.7$_{(-32.7)}$ | 24.5$_{(-36.0)}$ | 14.3$_{(-12.4)}$ | 19.4$_{(-73.2)}$ | 80.7$_{(-14.1)}$ |
| CLIP | ✗ | 81.8 | 35.6 | 63.5 | 29.3 | 53.2 | 21.9 | 35.5 | 17.9 | 20.8 | 86.7 |
| | ✓ | 78.2$_{(-12.4)}$ | 28.2$_{(-30.9)}$ | 61.3$_{(-14.3)}$ | 27.2$_{(-17.9)}$ | 45.9$_{(-34.0)}$ | 17.8$_{(-27.6)}$ | 32.9$_{(-27.6)}$ | 16.5$_{(-10.2)}$ | 20.2$_{(-72.4)}$ | 84.7$_{(-10.1)}$ |
| **MIM with Input Diversity (DIM) (Xie et al., 2019)** | | | | | | | | | | | |
| Deit-T | ✗ | 75.3 | 28.2 | 59.2 | 24.3 | 41.3 | 16.5 | 29.5 | 16.3 | 20.3 | 84.5 |
| | ✓ | 62.6$_{(-28.0)}$ | 16.9$_{(-42.2)}$ | 48.1$_{(-27.5)}$ | 20.5$_{(-24.6)}$ | 37.8$_{(-42.1)}$ | 13.4$_{(-32.0)}$ | 24.2$_{(-36.3)}$ | 14.3$_{(-12.4)}$ | 19.6$_{(-73.0)}$ | 85.0$_{(-9.8)}$ |
| Deit-S | ✗ | 76.3 | 29.4 | 57.5 | 24.1 | 39.8 | 17.3 | 29.8 | 16.2 | 21.8 | 85.9 |
| | ✓ | 54.1$_{(-36.5)}$ | 12.3$_{(-46.8)}$ | 42.8$_{(-32.8)}$ | 17.7$_{(-27.4)}$ | 36.8$_{(-43.1)}$ | 12.7$_{(-32.7)}$ | 22.3$_{(-38.2)}$ | 14.6$_{(-12.1)}$ | 21.0$_{(-71.6)}$ | 85.7$_{(-9.1)}$ |
| Deit-B | ✗ | 75.9 | 28.8 | 57.8 | 24.5 | 41.5 | 16.5 | 30.5 | 15.6 | 22.2 | 85.1 |
| | ✓ | 55.5$_{(-35.1)}$ | 12.6$_{(-46.5)}$ | 40.6$_{(-35.0)}$ | 16.8$_{(-28.3)}$ | 37.3$_{(-42.6)}$ | 11.4$_{(-34.0)}$ | 21.5$_{(-39.0)}$ | 13.3$_{(-13.4)}$ | 20.5$_{(-72.1)}$ | 86.3$_{(-8.5)}$ |
| DINO | ✗ | 64.9 | 23.1 | 51.8 | 22.6 | 43.5 | 16.7 | 27.6 | 13.7 | 19.0 | 80.2 |
| | ✓ | 45.2$_{(-45.4)}$ | 7.30$_{(-51.8)}$ | 35.8$_{(-39.8)}$ | 15.8$_{(-29.3)}$ | 32.2$_{(-47.7)}$ | 10.6$_{(-34.8)}$ | 18.3$_{(-42.2)}$ | 12.0$_{(-14.7)}$ | 18.9$_{(-73.7)}$ | 78.6$_{(-16.2)}$ |
| CLIP | ✗ | 78.0 | 29.5 | 61.0 | 26.9 | 50.0 | 19.6 | 32.8 | 16.9 | 20.8 | 87.8 |
| | ✓ | 73.4$_{(-17.2)}$ | 22.7$_{(-36.4)}$ | 56.7$_{(-18.9)}$ | 25.1$_{(-20.0)}$ | 46.5$_{(-33.4)}$ | 13.6$_{(-31.8)}$ | 31.5$_{(-29.0)}$ | 15.7$_{(-11.0)}$ | 20.5$_{(-72.1)}$ | 87.4$_{(-7.4)}$ |

*(Top section header: Momentum Iterative Fast Gradient Sign Method (MIM) (Dong et al., 2018))*

Table 2: ***Adversarial Transferability from image to video models:*** Adversarial accuracy (%) is reported at $\epsilon \leq 16$. When attack is optimized from an Image model without temporal prompts (✗), then we minimize the unsupervised adversarial objective (Eq. 3), cosine similarity, between the feature embedding of spatial class token of clean and adversarial frames. We use 8 frames from a given video. Attacks transferability improves by a clear margin with temporal prompts. Clean accuracy (top1 % on randomly selected validation set of 1.5k samples) is highlighted with cell color. *We observe that a small Image model (e.g. Deit-T with only 5 Million parameters) can significantly reduce the generalization of TimesFormer with the presence of our temporal prompts. The attack performance with temporal prompts increases with network capacity (e.g. Deit-T to Deit-B).*

to the black-box video model, its training data, or label space ( Appendix B). The maximum pixel perturbation is set to $\epsilon \leq 16$ for pixel range of [0, 255]. We use standard image attacks to optimize adversarial examples through our adapted image models and boost adversarial transferability to video models by using the following protocols:

**Surrogate image models:** We optimize adversaries from the known surrogate models. We study three types of surrogate image models trained via supervised (Touvron et al., 2020), self-supervised (Caron et al., 2021) and text-supervised (Radford et al., 2021) models.

$-$*Supervised image models:* We use Deit-T, DeiT-S, and DeiT-B with 5, 22, and 86 million parameters to video datasets. These models are pre-trained in a supervised manner on ImageNet.

$-$*Self-Supervised image model:* We use Vision Transformer trained in self-supervised fashion on ImageNet using DINO training framework (Caron et al., 2021).

$-$*Text-Supervised image model:* We adapt CLIP image encoder only trained on text-guided images.

DeiT-B, DINO, and CLIP share the same network based on Vision Transformer. These models process images of size $3 \times 224 \times 224$ that are divided into 196 patches with a patch size of 16. Each of these patch tokens has 768 embedding dimensions. Thus, the only difference between these models lies in their corresponding training frameworks. These models are adapted to different video datasets. Adversarial examples are then simply created using existing single and multi-step attacks (Goodfellow et al., 2014; Madry et al., 2018; Dong et al., 2018; Xie et al., 2019).

**Target video and image models:** We transfer adversarial examples to unknown (black-box) video and image models. $-$*Target Video Models:* We consider recently proposed TimesFormer (Bertasius et al., 2021) with divided space-time attention, 3D convolutional network (Tran et al., 2018), and multi-view convolutional network (Su et al., 2015). $-$*Target Image Models:* We consider the same image models used in baseline by (Naseer et al., 2022b): BiT-ResNet50 (BiT50) (Beyer et al.,

| MIM with Input Diversity (DIM) (Xie et al., 2019) | | | | | | | | | |
|---|---|---|---|---|---|---|---|---|---|
| | | Convolutional | | | | Transformer | | | |
| Models (↓) | Method | BiT50 | Res152 | WRN | DN201 | ViT-L | T2T24 | TnT | T2T-7 |
| Deit-B | Naseer et al. (2022b) | 80.10 | 84.92 | 86.36 | 89.24 | 78.90 | 84.00 | 92.28 | 93.42 |
| Deit-B | | **86.64** | 90.88 | 92.14 | 93.82 | 95.64 | **95.74** | **98.48** | 94.74 |
| DINO | Ours | 84.36 | **93.74** | **95.16** | **96.20** | **96.48** | 89.68 | 94.74 | **96.18** |
| CLIP | | 55.34 | 64.24 | 65.76 | 71.78 | 59.94 | 54.66 | 62.72 | 75.80 |

Table 3: ***Adversarial Transferability from image to image models:*** Fool Rate (%) on 5k ImageNet val. samples from Naseer et al. (2022b). We compare against the best results from Naseer et al. (2022b) that our baseline is a self-ensemble Deit-B with token refinement blocks. To mimic the dynamics with static images, we optimize our proposed prompts at resolutions of 56×56, 96×94, 120×120, and 224×224. Adversaries are optimized at the target resolution of 224×224. Our method performs favorably well against self-ensemble with refined tokens.

| Models | Temporal Prompts | SR | TimesFormer | |
|---|---|---|---|---|
| | | | UCF | HMDB |
| | ✗ | ✗ | 75.9 | 28.8 |
| DeiT-B | ✓ | ✗ | 55.5 | 12.6 |
| | ✓ | ✓ | **53.9** | **11.8** |

Table 4: ***Effect of Change in Spatial Resolution (SR):*** Incorporating change in spatial resolution improves adversarial transferability (*% accuracy, lower is better*). Adversaries are created at target resolution of 224 from an ensemble of models adopted at 56×56, 96×96, and 224×224.

2021), ResNet152 (Res152) (He et al., 2016), Wide-ResNet-50-2 (WRN) (Zagoruyko & Komodakis, 2016), DenseNet201 (DN201) (Huang et al., 2017) and other ViT models including Token-to-Token transformer (T2T) (Yuan et al., 2021), Transformer in Transformer (TnT) (Mao et al., 2021).

**Adapting image models for videos using dynamic cues:** We use UCF (Soomro et al., 2012), HMDB (Kuehne et al., 2011), K400 (Kay et al., 2017), and SSv2 (Goyal et al., 2017) training sets to learn temporal prompts and adapt image models to videos via our approach (Fig.1). HMDB has the smallest validation set of 1.5k samples. For evaluating robustness, we selected all validation samples in HMDB, while randomly selected 1.5k samples from UCF, K400, and SSv2 validation sets. We also use multi-view training samples rendered for 3D ModelNet40 (depth and shaded) for image models. We use validation samples of rendered multi-views for both modalities.

**Adapting image models to images mimicking dynamic cues:** We use ImageNet training set and learn our proposed transformation and prompts at multiple spatial scales; 56×56, 96×96, 120×120, and 224×224. We optimize adversarial attacks at the target resolution of 224×224 by using the ensemble of our learned models at multiple resolutions. In this manner, our approach mimic the change over time by changing the spatial scale or resolution over time. We study our attack approach using the 5k samples from ImageNet validation set proposed by (Naseer et al., 2022b).

**Inference and metrics:** We randomly sample 8 frames from a given video for testing. We report drop in top-1 (%) accuracy on adversarial and clean videos. We report fooling rate (%) of adversarial samples for which the predicted label is flipped w.r.t the original labels) to evaluate on image datasets.

**Extending image attacks to videos:** We apply image attacks such as single step fast gradient sign method (FGSM) (Goodfellow et al., 2014) as well as iterative attacks with twenty iterations including PGD (Madry et al., 2018), MIM (Dong et al., 2018) and input diversity (augmentations to the inputs) (DIM) (Xie et al., 2019) attacks to adapted image models and transfer their adversarial perturbations to black-box video models. We follow the attack settings used by (Naseer et al., 2022b). Our proposed transformation ($\mathcal{T}(.)$) allows to model temporal gradients during attack optimization (Fig. 3). We simply aggregates the gradients from both branches of our model to the input video frames (Fig. 1).

### 3.1 Adversarial Transfer from Image-to-Video Models

The generalization of image models on videos is discussed in Table 5. We observe that image solution remains preserved in ViT backbones even after training with our proposed transformation and temporal prompts. Deit models retain their top-1 (%) accuracy on ImageNet when measured through image class token, while also exhibiting decent performance on video datasets and rendered mulit-views of ModelNet40. CLIP achieves the best performance on videos. The transferability of image based attacks from our adapted image models to video and multi-view models is presented in Tables 1 and 2. Following insights emerges from our analysis: **a)** The attack success to video models increases with size of the image models. The larger the image model (*e.g.* Deit-T to DeiT-B) the

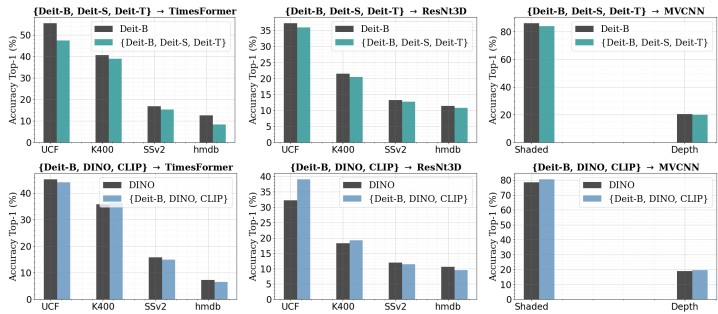

Figure 4: Adversaries (DIM (Xie et al., 2019)) from an ensemble of image models with different networks transfer well and fool the black-box video models. *First row* shows transferablity from an ensemble of different networks, while *second row* shows attack results from an ensemble of similar networks but pre-trained with different training schemes.

| Models | ImageNet | ImageNet and Videos | | | | ImageNet and ModelNet40 | |
|--------|----------|---------------------|---------|---------|---------|------------------------|---------|
|        | IN       | IN – UCF            | IN – HMDB | IN – K400 | IN – SSv2 | IN – Depth            | IN – Shaded |
| Deit-T | 72.2     | 72.0 − 70.0         | 72.3 − 36.2 | 72.2 − 44.6 | 72.2 − 11.2 | 72.2 − 86.0          | 72.1 − 81.0 |
| Deit-S | 79.9     | 79.8 − 77.2         | 79.8 − 44.6 | 79.9 − 53.0 | 79.9 − 15.3 | 78.8 − 86.6          | 79.8 − 86.2 |
| Deit-B | 81.8     | 81.9 − 81.4         | 81.7 − 47.7 | 81.9 − 57.0 | 81.8 − 17.5 | 81.8 − 90.1          | 81.4 − 88.2 |
| DINO   | NA       | NA − 79.5           | NA − 45.1 | NA − 57.4 | NA − 17.4 | NA − **90.1**        | NA− **89.8** |
| CLIP   | NA       | NA − **86.0**       | NA − **54.6** | NA − **67.3** | NA − **19.9** | NA − 89.5          | NA − 88.9 |

Table 5: *Spatial along with Temporal Solutions within Image Models:* Our approach successfully incorporates temporal information into pre-trained and frozen image models and increases their generalization (top-1 (%) at resolution 224) to high-dimensional datasets. The original image solution remains preserved in our approach. Ours Deit-B retains 81.9% top-1 accuracy on ImageNet validation set while exhibiting 81.4% on UCF.

higher the attack success, **b)** The adversarial perturbations generated using self-supervised DINO transfer better than CLIP or supervised Deit-B, **c)** Divided space-time attention is more robust than 3D convolution, and **d)** MVCNN trained on shaded rendered views is more robust than depth.

### 3.1.1 ENSEMBLE ADVERSARIAL TRANSFER

Adversarial attacks optimized from an ensemble of models (Dong et al., 2018; Xie et al., 2019) or self-ensemble (Naseer et al., 2022b) increases transferability. We study three types of image ensembles including **a)** *Ensemble of Different Networks:* We transfer attack from three image models of the Deit family (Touvron et al., 2020) including Deit-T, Deit-S and Deit-B. These models differ in architecture, **b)** *Ensemble of Different Training Frameworks:* We transfer attack from three image models including Deit-B, DINO and CLIP. These models share similar architecture but differ in their training, and **c)** *Ensemble of Different Spatial Resolutions:* We transfer attack from three image models with the same architecture (Deit-B) but prompt adapted to different spatial resolutions.

Ensemble of different training frameworks performs favorably to boost attack transferability (Fig. 4). An attacker can adapt our approach at different resolutions to enhance transferability (Table 4). We provide generalization of Deit-B at varying spatial scales for video datasets in Appendix I.

### 3.2 ADVERSARIAL TRANSFER FROM IMAGE-TO-IMAGE MODELS

Image datasets are static but we mimic the dynamic behaviour by learning our proposed transformation at different spatial scales (Fig. 2). Our approach exhibit significant gains in adversarial transferability (Table 3). Refer to appendices B-J for extensive analysis on cross-task adversarial transferability, textual bias of CLIP on adversarial transferability, visualization of attention roll-outs and latent embeddings, and effect of the number of temporal prompts on adversarial transferability.

## 4 CONCLUSION

In this work, we study the adversarial transferability between models trained for different data modalities (*e.g.*, images to videos). We propose a novel approach to introduce dynamic cues within pre-trained (supervised or self-supervised) image models by learning a simple transformation on videos. Our transformation enables temporal gradients from image models and boost transferability through temporal prompts. We show how our approach can be extended to multi-view information from 3D objects and varying image resolutions. Our work brings attention to the image models as universal surrogates to optimize and transfer adversarial attacks to black-box models.

**Reproducibility Statement**: `Training image models`: We adapt pre-trained image models Deit, DINO and CLIP, open sourced by the corresponding authors. We freeze the weights of these models and adapt them for video datasets by introducing our modifications (transformation with temporal prompts). We incorporate the adapted models into the open sourced TimesFormer github repo `https://github.com/facebookresearch/TimeSformer` and train them with default settings. `Training video models`: We use the TimesFormer github repo with default settings to train ResNet3D on UCF, HMDB, K-400, and SSv2. `Training MVCNN`: We use open source code `https://github.com/jongchyisu/mvcnn_pytorch` to train multi-view models on rendered views (depth, shaded) of ModelNet40. `Attacks`: We use adversarial attacks with default parameters as described in baseline (Naseer et al., 2022b). `Datasets`: We use the same 5k image samples from ImageNet validation set, open sourced by Naseer et al. (2022b), while randomly selected 1.5k samples from the validation sets of explored video datasets. Furthermore, we will release the pre-trained model publicly.

**Ethics Statement**: Our work highlights the use of freely available pre-trained image models to generate adversarial attacks. This can be used to create attacks on real-world deep learning systems in the short run. Our work also provides insights on how such attack can emerge from different resources. Therefore, our work provides an opportunity to develop more robust models to defend against such attacks. Further, the image and video datasets used in this work also pose privacy risks which is an important topic we hope to explore in future.

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

# Appendix

Our approach simply augments the existing adversarial attacks developed for image models and enhances their transferability to black-box video models using our proposed temporal cues. We study cross-task adversarial transferability ((Naseer et al., 2019)); the attacker has no access to the black-box video model, its training data, or label space in Appendix B. We compare the adversarial transferability of our best image models (DINO (Caron et al., 2021)) with temporal cues against pure video models (TimesFormer Bertasius et al. (2021), ResNet3D (Tran et al., 2018), and I3D (Carreira & Zisserman, 2017) in Appendix C. We observe that adversarial examples found within adapted CLIP models show lower transferability to image and video models (Tables 1, 2, and 3). We analyze the textual biases of CLIP vision features in Appendix D to further explain the difference between adversarial space of uni-modal and vision-language models such as CLIP. In Appendix E, we analyze the attention roll-out from our temporal prompts to signify that our prompts pay attention to temporal information across frames as compared to the original image features. We also observe the behavior of our temporal prompts across shuffled frames in Appendix F. We study the effect of increasing the number of temporal prompts during the attack in Appendix G. The effect of fully fine-tuning the image models with our proposed temporal prompts on attack transferability is presented in Appendix H. The generalization of the image model (Deit-B) at varying spatial scales on video modalities is presented in appendix I. In appendix J, we visualize the latent space of class tokens trained at a spatial resolution of $224 \times 224$ vs our adapted tokens at various spatial resolution scales. The evolution of adversarial patterns across time generated using our approach is provided in appendix K.

## A    RELATED WORK

**Adversarial Attacks:** Transferable adversarial attacks compromise the decision of a black-box target model. Attacker can not access weights, model outputs or gradient information of a black-box model. These attacks can broadly be categorized into iterative (Goodfellow et al., 2014; Dong et al., 2018; Lin et al., 2019; Wang & He, 2021; Xie et al., 2019; Zhou et al., 2018; Naseer et al., 2022b; Wei et al., 2022) and generative (Poursaeed et al., 2018; Naseer et al., 2019; 2021; 2022a) methods. Iterative attacks optimize adversary from a given clean sample by searching adversarial space of a white-box surrogate model iteratively. Attacker can access weights, model outputs and gradient information of a white-box surrogate model. Generative attacks on the other hand train a universal function (an auto-encoder) against the white-box model over a given data distribution. These attacks can generate adversary via single forward-pass after training. Our proposed approach can be augmented with any of these existing attacks to boost transferability from image-to-video or multi-view models.

**Prompt Adaptation of Image Models:** Prompting refers to providing manual instructions (Liu et al., 2021a; Jiang et al., 2020; Shin et al., 2020) or fine-tuning task-specific tokens (Li & Liang, 2021; Liu et al., 2021b; Lester et al., 2021) for a desired behavior from large language models (*e.g.*, GPT-3). Recently, prompt fine-tuning has been studied for vision models with promising results to adapt image models from one image task to another (Jia et al., 2022; Elsayed et al., 2018). We extend prompt transfer learning to adapt image models to videos and multi-view data modalities. We only train a transformation along with a prompt class token to learn dynamic cues and adapt image model to video tasks. Our approach allows transferable attacks from image-to-video or multi-view models.

## B    DYNAMIC CUES TO BOOST CROSS-TASK TRANSFERABILITY

We apply the cross-task attack using self-supervised objective (cosine similarity such as described in Eq. 3 in Sec. 2.2) to our surrogate image models with temporal cues and compare the adversarial transferability with pure video surrogate models (I3D (Carreira & Zisserman, 2017), and Timesformer (Bertasius et al., 2021)) in Table 6. The cross-task attack transferability when combined with our approach described in Section 2.2 performs favorably well against the video models. This is contributed to the fact that the attack exploits an ensemble of two different representations (image and video) when combined with our adapted image models. Therefore our approach frees the attacker from having access to a specialized video architecture trained specifically on the video to generate more transferable attacks.

| MIM with Input Diversity (DIM) (Xie et al., 2019) | | | | | | | | |
|---|---|---|---|---|---|---|---|---|
| | | | UCF | | | HMDB | | |
| Source Domain | Model (↓) | Temporal Prompts | I3D | TimesFormer | ResNet3D | I3D | TimesFormer | ResNet3D |
| K400 | TimesFormer | – | 35.0 | – | 42.1 | 21.3 | – | 21.1 |
| | I3D | – | – | 75.6 | 47.9 | – | 25.1 | 19.0 |
| | DINO (Ours) | ✓ | **13.2** | **63.2** | **41.5** | **8.4** | **22.9** | **15.8** |
| SSv2 | TimesFormer | – | 38.2 | – | **40.6** | 22.6 | – | 21.0 |
| | I3D | – | – | 78.0 | 45.0 | – | 29.1 | 18.9 |
| | DINO (Ours) | ✓ | **18.7** | **62.9** | 42.6 | **10.9** | **21.6** | **16.1** |

Table 6: ***Cross-Task attack with Temporal Prompts:*** Adversarial attack using the self-supervised image model (Caron et al., 2021) with our dynamic cues (Section 2.2) performs favorably as compared to pure video models.

| Fast Gradient Sign Method (FGSM) (Goodfellow et al., 2014) | | | | | |
|---|---|---|---|---|---|
| | | UCF | | HMDB | |
| Model (↓) | Temporal Prompts | TimesFormer | ResNet3D | TimesFormer | ResNet3D |
| TimesFormer | – | – | 30.1 | – | 15.6 |
| ResNet3D | – | 69.9 | – | 30.7 | – |
| DINO (Ours) | ✓ | **60.7** | **21.9** | **18.8** | **8.3** |
| **Projected Gradient Decent (PGD) (Madry et al., 2018)** | | | | | |
| TimesFormer | – | – | 73.3 | – | 36.3 |
| ResNet3D | – | 86.0 | – | 46.5 | – |
| DINO (Ours) | ✓ | **64.9** | **63.1** | **14.8** | **25.9** |
| **Momentum Iterative Fast Gradient Sign Method (MIM) (Dong et al., 2018)** | | | | | |
| TimesFormer | – | – | 44.5 | – | 21.5 |
| ResNet3D | – | 77.4 | – | 35.0 | – |
| DINO (Ours) | ✓ | **47.7** | **39.8** | **8.40** | **12.7** |
| **MIM with Input Diversity (DIM) (Xie et al., 2019)** | | | | | |
| TimesFormer | – | – | 41.7 | – | 18.4 |
| ResNet3D | – | 74.4 | – | 34.4 | – |
| DINO (Ours) | ✓ | **45.2** | **32.2** | **7.30** | **10.6** |

Table 7: ***Temporal Prompts Vs. Video Models:*** Adversarial attack using the self-supervised image model (Caron et al., 2021) with our dynamic cues (Section 2.2) performs favorably as compared to pure video models.

| Generalization of Video Models | | | |
|---|---|---|---|
| Dataset (↓) | TimesFormer | ResNet3D | I3D |
| UCF | 90.6 | 79.9 | 79.7 |
| HMDB | 59.1 | 45.4 | 43.8 |
| K400 | 75.6 | 60.5 | 73.5 |
| SSv2 | 45.1 | 26.7 | 48.4 |

Table 8: ***Generalization of Video Models:*** The top-1 (% ) accuracies on the validation datasets are reported. We provide single view inference on 8 frames to be consistent across these video models.

## C   IMAGE VS. VIDEO SURROGATE MODELS

The attack transferability of our image models with dynamic cues is compared against pure video models (TimesFormer (Bertasius et al., 2021), ResNet3D (Tran et al., 2018), and I3D (Carreira & Zisserman, 2017)) in Table 7. The generalization (top-1 % accuracies) of the video models (Table 8) is higher than our adapted image models (Table 5). This is expected as video models are specifically designed and trained from scratch on the video datasets. However, the adversarial transferability of our image models compares well against these pure video models (Table 7). Our approach enables the attacker to exploit image and video solution from the same image model while pure video models lack the image representation.

## D   ANALYSING CLIP

Vision-language models like CLIP consists of an image and a text encoder and are trained on large-scale image-text pairs to find common feature space between images and textual labels. Inherently visual features of CLIP are aliened with text. Our results indicate that adversarial perturbations

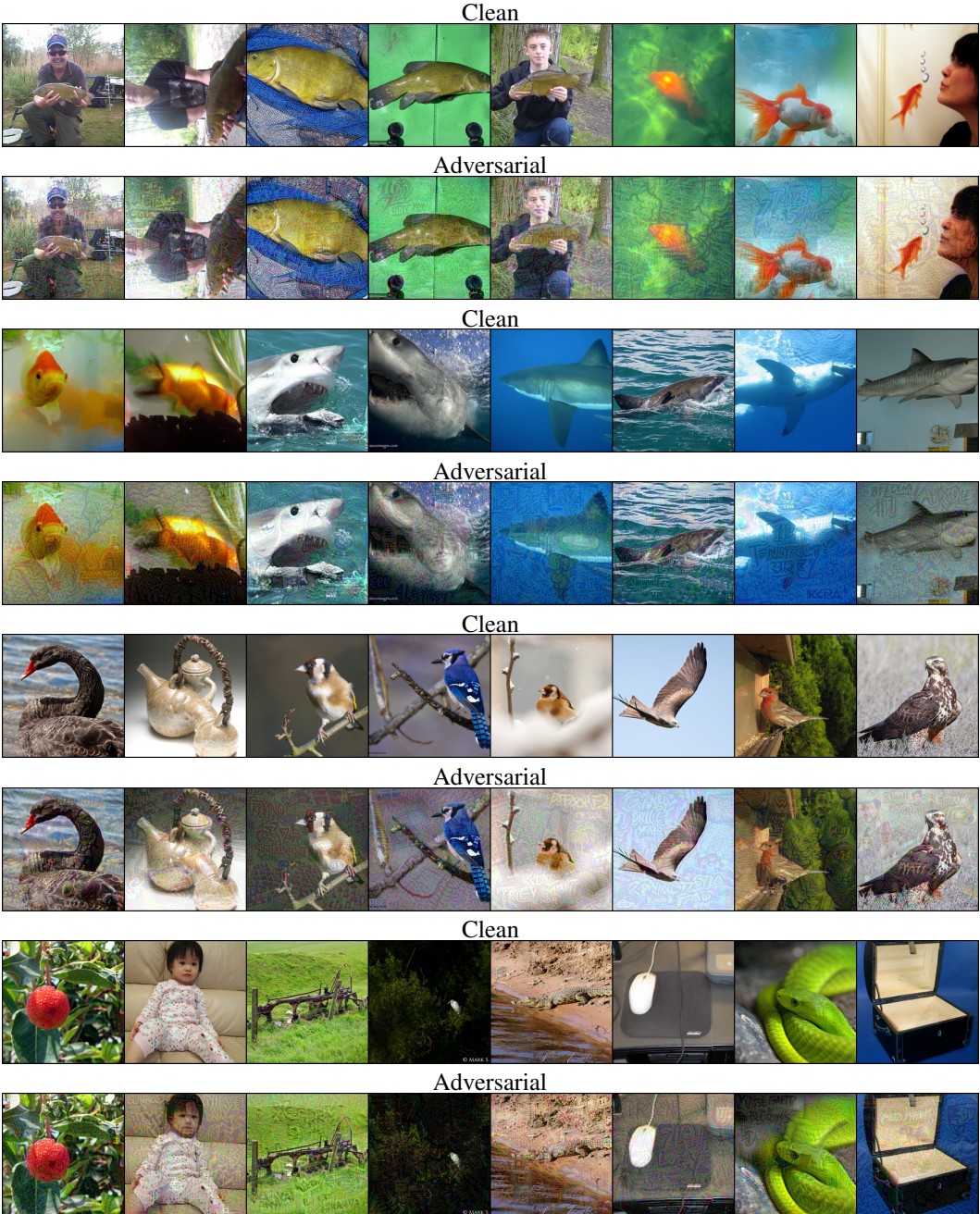

Figure 5: *Visualizing Textual Bias of CLIP Adversarial Space:* We apply DIM attack (Xie et al., 2019) on the original CLIP model. It generates adversaries with text imprints. These perturbation successfully fools CLIP but does not transfer well on image models since the presence of textual bias is not optimal for black-box image models. We adapt the pre-trained and frozen CLIP vision encoder to videos. Therefore, it is expected that after training our temporal prompts introduced within CLIP also contain textual bias as these prompts interact with CLIP frozen features. *Best Viewed in Zoom.*

found via the CLIP vision encoder show lower transferability as compared to similar models such as DINO (Caron et al., 2021) and Deit-B (Touvron et al., 2020). The alignment of CLIP vision features towards text results into adversaries with textual bias. We can see textual patterns emerging within adversarial perturbations (Fig. 5). These perturbations are optimal to fool CLIP but not that meaningful for vision models trained without text supervision. We quantify this as well by measuring cosine similarly between visual features of CLIP vision encoder, DINO, and Deit-B with CLIP

| Measuring Textual Bias | | |
|---|---|---|
| Vision Encoder | Text Encoder | Cosine Similarity |
| CLIP | CLIP | 0.31 |
| Deit-B | CLIP | 0.005 |
| DINO | CLIP | 0.002 |

Table 9: *Textual Bias of Vision Encoders:* Results are reported on 1000 randomly selected images (one per class) from ImageNet validation set. We measure the similarity between the refined class token output by vision encoders with textual features output by CLIP text encoder for text label input as "a photo of a {class name}".

| Inference on Shuffled Frames | | | | | |
|---|---|---|---|---|---|
| Model (↓) | Temporal Prompts | UCF | HMDB | k400 | SSv2 |
| Deit-B | ✓ | 80.31±0.33 | 46.2±0.22 | 57.1±0.35 | 16.9±0.52 |
| DINO | ✓ | 78.40±0.35 | 46.1±0.19 | 55.7±0.21 | 17.1±0.32 |
| CLIP | ✓ | 85.6 ± 0.23 | 55.0 ± 0.18 | 66.2 ± 0.33 | 19.1 ± 0.21 |

Table 10: *Robustness to Temporal Variations:* We repeat inference results three times on a single view with 8 randomly sampled and shuffled frames. Our adapted image models show higher robustness to such temporal variations.

| Effect of Number of Temporal Prompts on Transferability | | | | | | | | | |
|---|---|---|---|---|---|---|---|---|---|
| | | TimesFormer | | | | ResNet3D | | | |
| Model (↓) | Temporal Prompts | UCF | HMDB | K400 | SSv2 | UCF | HMDB | K400 | SSv2 |
| | 8 | 55.5 | 12.6 | 40.6 | 16.8 | 37.3 | 11.4 | 21.5 | 13.3 |
| Deit-B | 16 | 51.4 | 10.6 | 38.7 | 15.3 | 35.2 | 9.9 | 20.6 | 15.5 |
| | 32 | **21.5** | **6.5** | **23.1** | **10.4** | **19.6** | **6.0** | **13.9** | **11.2** |
| | 8 | 45.2 | 7.3 | 35.8 | 15.8 | 32.2 | 10.6 | 18.3 | 12.0 |
| DINO | 16 | 42.9 | 6.0 | 35.3 | 14.5 | 29.7 | 8.8 | 18.1 | 11.5 |
| | 32 | **30.2** | **5.7** | **22.6** | **8.3** | **15.3** | **4.9** | **10.0** | **7.3** |

Table 11: *Effect of Number of Temporal Prompts on Adversarial Transferability:* We observe non-trivial gains in adversarial transferability as we increase our temporal prompts. This indicates that the more the temporal information the better the attack transferability (DIM (Xie et al., 2019)) of our approach (*lower is better*).

textual features produced for text labels (Table 9). These observations further explain the lower performance of the ensemble of different training frameworks ({DINO, CLIP, Deit-B}) as compared to an ensemble of different networks ({Deit-T, Deit-S, Deit-B}) (Fig. 4).

# E   ATTENTION ROLL-OUT ACROSS TIME

We compute the roll-out attention (Abnar & Zuidema, 2020) of our temporal prompts within image models and compared those against original image models (Fig. 6). We observe that our temporal prompts focuses on action across different frames for a video as compared to the spatial class token with in pre-trained image models. Fooling such models allows for finding transferable adversarial patterns.

# F   ANALYSING DYNAMIC CUES AGAINST TEMPORAL VARIATIONS

We further analyse our dynamic cues by measure robustness to temporal variations. We provide inference results on shuffled video frames in Table 10. We randomly sample a single view with 8 frames and report average and standard deviation across three runs. Our models are robust to such temporal variations.

# G   EFFECT OF NUMBER OF TEMPORAL PROMPTS

It's a common practice to increase the number of frames for better temporal modeling. In similar spirit, we also increase number of training frames and consequently the number of our temporal prompts through transformation. As expected, this strategy leads to better performance in modeling dynamic cues and attack transferability (Table 11).

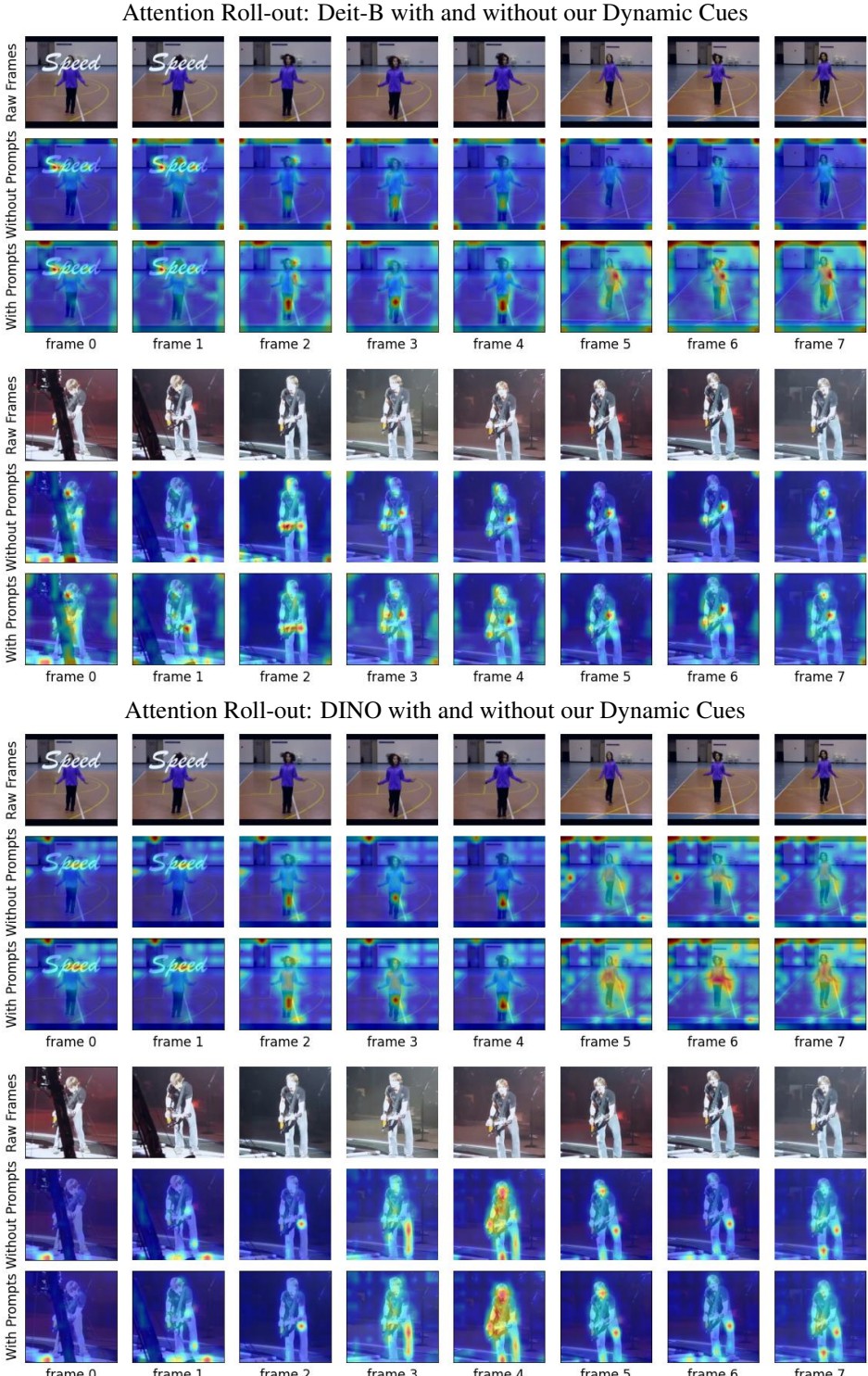

Figure 6: *Visualization of Dynamic Cues through Attention role-out:* We use (Abnar & Zuidema, 2020) to observe the most salient regions for the spatial and temporal class tokens. Our temporal class token pay more attention to action within a video across temporal frames.

| Temporal Prompting Vs. Fine-Tuning | | | | |
|---|---|---|---|---|
| | | | TimesFormer | |
| Models ($\downarrow$) | Temporal Prompts | Fine Tuning | UCF | HMDB |
| DINO | ✓ | ✓ | 56.3 | 13.6 |
| | ✓ | ✗ | **45.2** | **7.3** |
| Deit-B | ✓ | ✓ | 59.3 | 12.1 |
| | ✓ | ✗ | **55.5** | **12.6** |

Table 12: ***Temporal Prompts Vs. Full Fine Tuning:*** Our proposed temporal Prompts perform favorably as compared to the full fine-tuning of the image model for dynamic cues. Full fine-tuning affects the pre-trained image representations which in turn lowers the contribution of self-supervised attack loss (Eq. 3) during the attack, reducing adversarial transferability.

| Model | Temporal Prompts | Total Parameters (Millions) | Trainable Parameters (Millions) | GFLOPs |
|---|---|---|---|---|
| Deit-B | ✗ | 86.0 | – | 140.6 |
| Deit-B | ✓ | 95.0 | 7.7 | **34.1** |
| DINO | ✗ | 86.0 | – | 140.6 |
| DINO | ✓ | 95.0 | 7.7 | **34.1** |
| CLIP | ✗ | 85.6 | – | 139.5 |
| CLIP | ✓ | 93.6 | 7.4 | **34.1** |
| TimesFormer | – | 121.7 | 121.7 | 196 |

Table 13: Our approach has significantly less computational overhead with few learnable parameters. In comparison to video models such as TimesFormer, we need to train only 7 million parameters. Note that video models like Timesformer simultaneously forward pass all sampled frames through the model layers. The image models like Deit can only process one frame at a time so these models require multiple forward pass to process a video. In contrast, our single transformation layer process a video to generate temporal prompts which are combined with a randomly selected single frame to forward pass through our image models. As result, our approach is computationally cheaper than pure video or image models while increasing the adversarial transferability. We present these results for single view with 8 frames.

# H  PROMPT LEARNING VS. FINE TUNING

We aim to use the same image backbone for different video or multi-view datasets. It has lower memory cost and computational overhead as we don't need to update all the model parameters during the backpass (Table 13).

Our attack exploits pre-trained image representations (for example learned on ImageNet) along with dynamic cues (introduced via learning temporal prompts) to fool black-box video models. These adapted image models are not optimal for video learning as compared to specialized video networks such as TimesFormer (Bertasius et al., 2021). When we fully fine-tune the pre-trained image model with our design to videos, the performance of the adapted models improves, in terms of top-1 (%) accuracy, on video datasets. As a result of this fine-tuning, however, we loss the original image representations otherwise preserved in our approach. This means that the contribution of our self-supervised loss ($\mathcal{L}_{ss}$), designed to exploit image level representation, reduces and as a result it affects the transferability of adversarial attacks.

Our prompt learning approach preserves the original image representations within the spatial class token which is used in our self-supervised loss to boost adversarial transferability. We lose the original image representations when doing full fine-tuning of our model on videos and as a result, transferability decreases ( Table 12 ). Interestingly this tradeoff between full fine-tuning and adversarial transferability is even more prominent with self-supervised DINO model as supervised fine-tuning affects the self-supervised inductive biases learned by the DINO.

In conclusion, attacks from pure image representations that is without temporal prompts, or from pure video representation that is without image representation show relatively lower transferability. The combination of adversarial image and video representations compliments each other in enhancing the transferability to black-box videos and image models. Both image and video representations are well preserved in our approach with in a single image model.

# I IMAGE MODELS FOR VIDEOS AT VARYING SPATIAL SCALES

We observe that our approach significantly improve generalization of image models on videos with low spatial resolutions (Table 14).

| Model | UCF | | | HMDB | | |
|-------|-----|-----|-----|------|-----|-----|
| | 224×224 | 96×96 | 56×56 | 224×224 | 96×96 | 56×56 |
| DeiT-B | 81.4 | 60.0 | 32.3 | 47.7 | 31.1 | 15.4 |

Table 14: ***Adapting image models to videos at varying spatial resolutions (SR):*** We showcase the generalization of DeiT-B model adopted for UCF and HMDB at multiple spatial resolutions.

# J VISUALIZING LATENT EMBEDDINGS

We show latent space embedding of our prompts for low resolution inputs in Fig. 7.

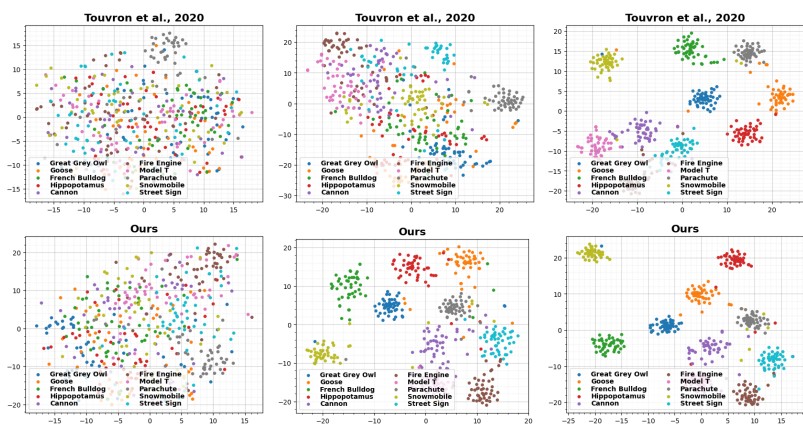

The class tokens of DeiT-S models projected from $\mathbb{R}^{384}$ to $\mathbb{R}^2$.

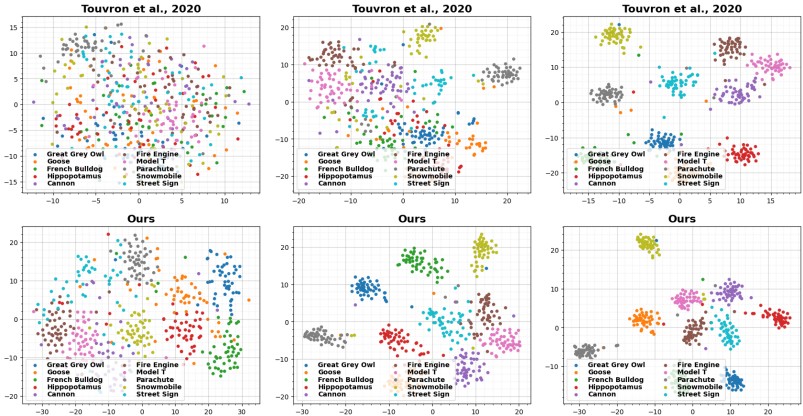

The class tokens of DeiT-B models projected from $\mathbb{R}^{768}$ to $\mathbb{R}^2$.

Figure 7: We adopted pre-trained Deit models (Touvron et al., 2020) at varying spatial resolutions using our approach; 56×56, 96×96, 120×120 ***(from left to right)***. We showcase inter-class separation and intraclass variation via t-SNE (Van der Maaten & Hinton, 2008) on 10 classes from ImageNet validation set. Our adopted image model have lower intraclass variations and better interclass separation than the original models. We used sklearn (Pedregosa et al., 2011) with perplexity set to 30 for all the experiments. *(best viewed in zoom)*.

# K VISUAL DEMOS

Figures 8, 9, and 10 shows adversarial pattern from our adapted image models, Deit-B, DINO, and CLIP, for different video samples.

Clean video sample

Adversarial patterns of DINO for a video sample.

Adversarial patterns of CLIP for a video sample.

Adversarial patterns of Deit-B for a video sample.

Figure 8: *Visualizing the behavior of Adversarial Patterns across Time:* We generate adversarial signals from DINO, CLIP, and DEIT-B models with temporal prompts using DIM attack (Xie et al., 2019). Observe the change in gradients across different frames (*best viewed in zoom*).

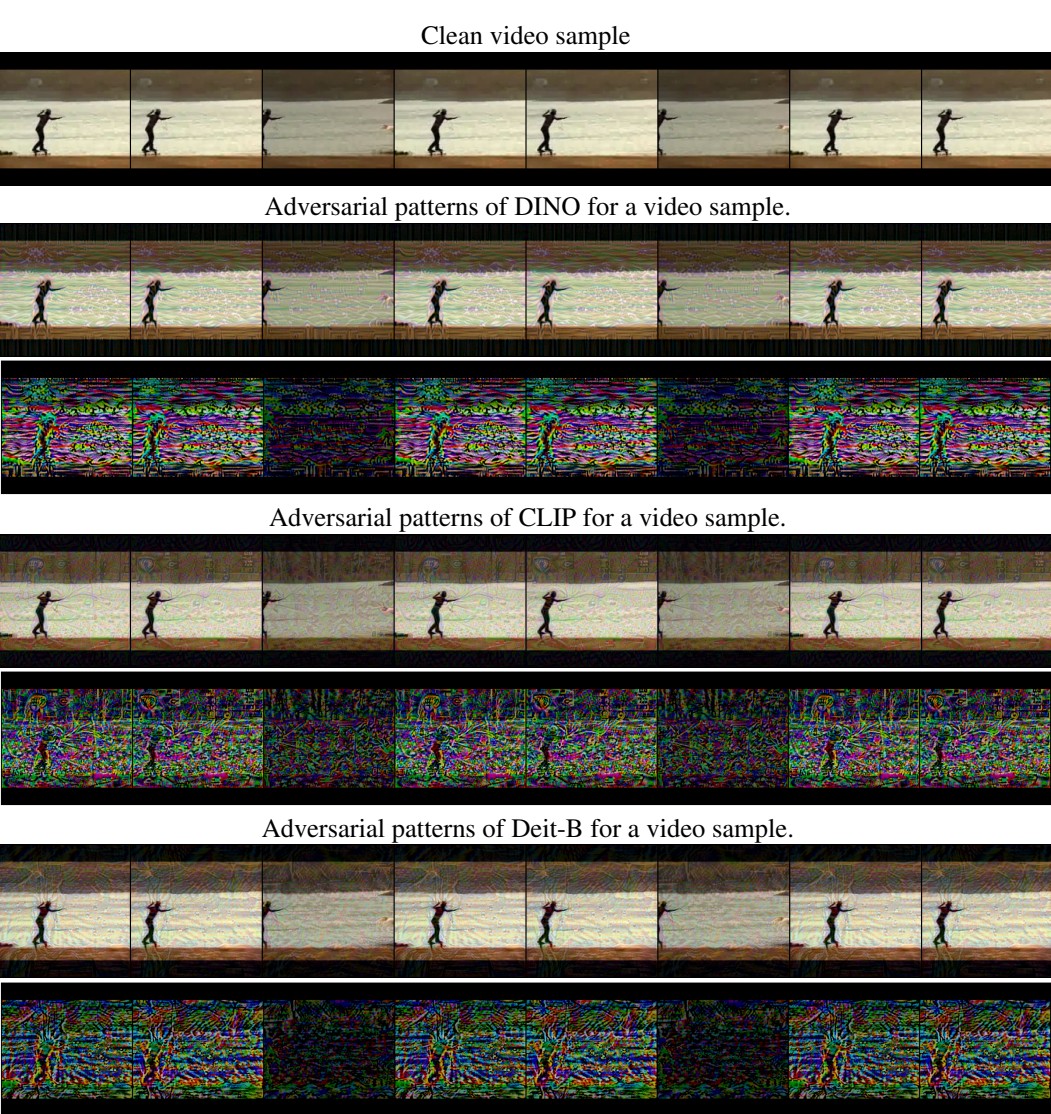

Figure 9: ***Visualizing the behavior of Adversarial Patterns across Time:*** We generate adversarial signals from DINO, CLIP, and Deit-B models with temporal prompts using DIM attack (Xie et al., 2019). Observe the change in gradients across different frames (*best viewed in zoom*).

Clean video sample

Adversarial patterns of DINO for a video sample.

Adversarial patterns of CLIP for a video sample.

Adversarial patterns of Deit-B for a video sample.

Figure 10: *Visualizing the behavior of Adversarial Patterns across Time:* We generate adversarial signals from DINO, CLIP, and Deit-B models with temporal prompts using DIM attack (Xie et al., 2019). Observe the change in gradients across different frames (*best viewed in zoom*).

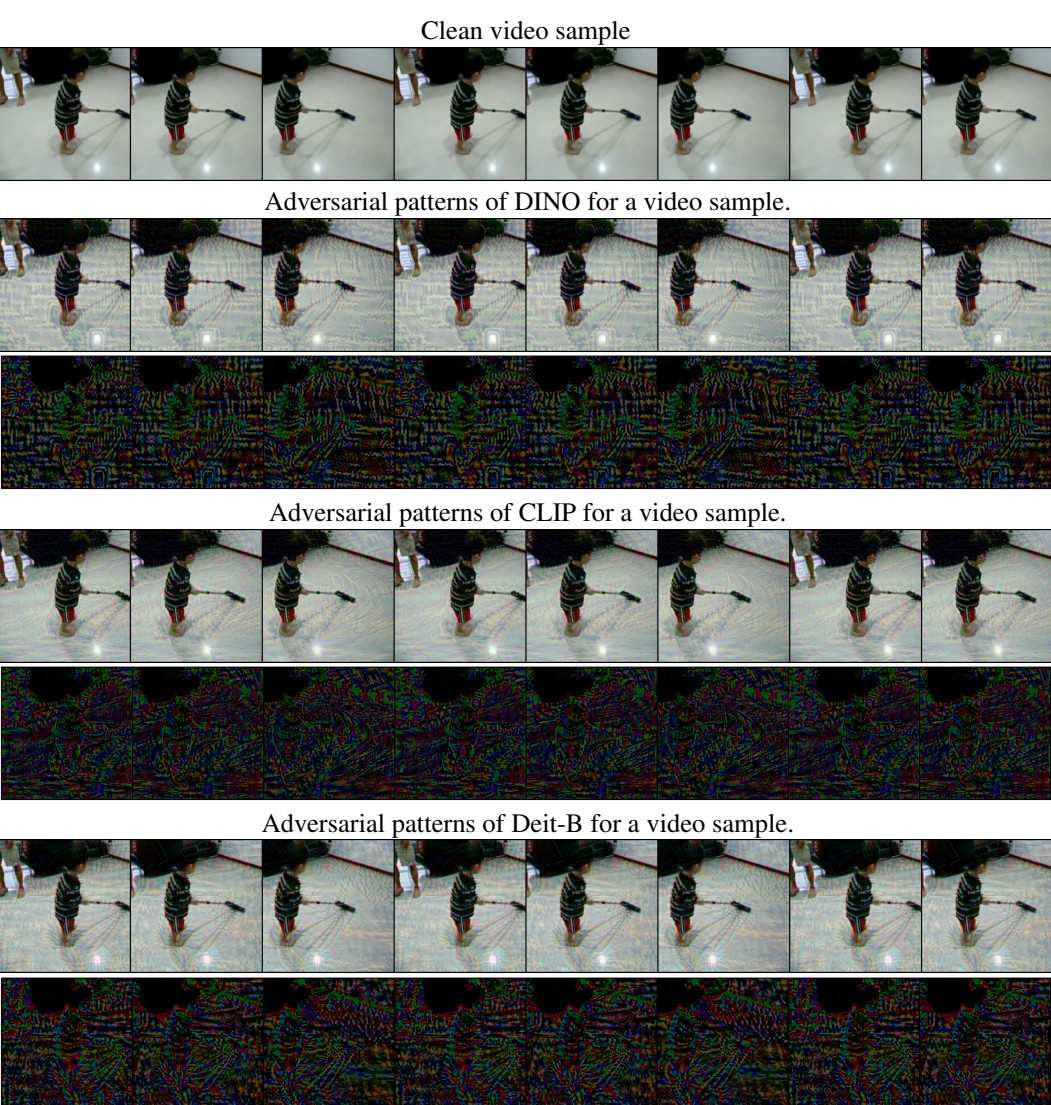

Figure 11: ***Visualizing the behavior of Adversarial Patterns across Time:*** We generate adversarial signals from DINO, CLIP, and Deit-B models with temporal prompts using DIM attack (Xie et al., 2019). Observe the change in gradients across different frames (*best viewed in zoom*).

