# OpenReview forum: "Boosting Adversarial Transferability using Dynamic Cues"
_ICLR.cc/2023/Conference — ICLR 2023 poster_

### Official Review · Reviewer_QXGV · 2022-10-25

**Confidence:** 3
**Correctness:** 3
**Technical Novelty And Significance:** 2
**Empirical Novelty And Significance:** 2
**Recommendation:** 6

**Clarity, Quality, Novelty And Reproducibility:**

### Clarity/Quality
- There are several run-on sentences and grammatical mistakes in the paper, which could be improved after revision. Some examples include: "The patch tokens combined with the class token1 cls ∈ R 1×D (are?) processed by multiple multi-head self-attention (MHSA) blocks before passing the cls through the task specific head MLP."

- Table 1 and Table 2 are not referenced explicitly in text.

### Novelty
- Applying prompt fine-tuning to adversarial transfer is novel.

Questions:
- Why is projected gradient descent (multi-step) weaker than FGSM for Table 1? (Does FGSM transfer better than PGD in general?) Also do they use different epsilons?


**Strength And Weaknesses:**

## Strength
- As far as I know, this is the first work that tries to transfer adversarial examples from image models to video models using prompt tuning.
- It seems their approach empirically works well.
- "Mimicking dynamic behavior on image datasets: Image samples are static and have no temporal dimension. So we adopt a simple strategy to mimic changing behavior for such static data. We consider images at different spatial scales to obtain a scale-space (Fig. 2) and model temporal prompts for image dataset"
	- This is equivalent to approximating camera motions via zooming. It's interesting that such a simple approach works.

## Weaknesses

- The way to present their results is sometimes confusing. For example, I think it would be easier to see if the authors only focus on Deit-B, Dino, and CLIP in the Table 1, and expand the discussion regarding the difference between training strategies. Separately, it would be nice to have more discussion regarding the model size difference. For example, I'm surprised that Deit-T, Deit-S and Deit-B are performing almost the same for FGSM in Table1. Can you comment on that?

- Section 4.1.1 (Ensemble Adversarial Transfer) needs more clarification. What attack was used to produce the numbers in Figure 4? It seems like for some datasets (e.g. SSv2, Shared, Depth, ) there is not benefit for ensemble adversarial transfer when I compare the bar plot of Figure 4 and the numbers of Table 2. If you claim that adversarial transfer is boosted by ensemble, I think the work should compare Figure 4 with the the number of Table 2. As of now, it's hard to see this.


**Summary Of The Paper:**

Adversarial transferability is often studied in the image-model-to-image-model setting. This paper studies how to transfer adversarial examples from image models to video (or multi-view) models. They note that simply transferring adversarial examples from image models to video models are suboptimal, and propose a prompt tuning method, inspired by recent development in prompt fine-tuning. They demonstrate that the proposed temporal prompt adaptation to learn dynamic cues on videos or multi-view renderings of 3D objects is effective to transfer adversary from pre-trained image models to video models and multi-view models.

**Summary Of The Review:**

While prompt fine-tuning has been studied before and this work is a straightforward application of prompt fine-tuning, applying prompt fine-tuning to adversarial transfer is novel, and it seems that their approach empirically works well.

---

> ### Author Response · Authors · 2022-11-19
> **Response to Reviewer  QXGV**
>
> We thank the reviewer for the encouraging and insightful comments. Please find our responses to specific queries below.
>
> **Model size difference and performance of FGSM.**
>
> Please note that FGSM is a single-step optimization attack. It's a weak attack baseline but is usually studied in adversarial transferability literature [1,2]. Our results are consistent with previous methods e.g [2] observe no gain in FGSM performance with the surrogate model size. Iterative (multi-step) attacks are proposed for better optimization to boost transferability [1].
>
> **Does FGSM transfer better than PGD in general?**
>
> PGD is an iterative (multi-step) attack. It overfits the surrogate models, a phenomenon well studied in adversarial literature [1], that's why different heuristics like momentum [1], and data augmentations [3,4] are proposed to avoid attack overfitting on the surrogate model and enhance transferability. Being a single step, FGSM is also a weak attack but it does not overfit the model and hence shows better performance than PGD [1].
>
> **Do they (attacks) use different epsilons?**
>
> All attacks studied in our paper use an epsilon of 16 within $l_\infty$ norm. We mention this detail in our experiments (Section 3).
>
> **What attack was used to produce the numbers in Figure 4?**
>
> Thank you for pointing it out. We use the DIM attack [3] to produce results in Figure 4. We mention this in the text highlighted in red.
>
> **Figure 4 and Benefits of ensemble adversarial transfer.**
>
> Our aim was to study different kinds of ensemble learning for adversarial transfer. We redraw the figure 4 and compare the ensemble results with the best-performing single model within that ensemble. Specifically, we compare different network ensemble {Deit-T, Seti-S, Deit-B} with Deit-B. Different training ensemble {Deit-B, DINO, CLIP} is compared against DINO. We observe that ensemble of different training performs poorly as compared to single model DINO mainly because of CLIP textual biases within adversarial perturbations (**please see Appendix D**). Please also see the response to Reviewer iNv7  “Analyses on CLIP’s low adversarial transferability”.
>
> The following table presents an analysis of textual bias within vision encoders studied in our work.
> | | | |
> | -- | :-: | :-:  |
> | Vision Encoder | Text Encoder | Cosine Similarity |
> | CLIP     | CLIP      | 0.31     |
> | Deit-B   | CLIP        | 0.005     |
> | DINO    | CLIP        | 0.002     |
>
> _Textual Bias of Vision Encoders: Results are reported on 1000 randomly selected images (one per class) from ImageNet validation set. We measure the similarity between the refined class token output by vision encoders with textual features output by CLIP text encoder for text label input as ``a photo of a \{class name\}"._
>
> **Minor Issues**
>
> Thank you for pointing these out. We have fixed these in our updated draft.
>
> **References**
>
> [1] Yinpeng Dong et.al “Boosting Adversarial Attacks with Momentum”, CVPR, 2018
>
> [2] Naseer et.al  “On Improving Adversarial Transferability of Vision Transformers ”, ICLR,2022
>
> [3] Cihang Xie et.al "Improving Transferability of Adversarial Examples with Input Diversity", CVPR, 2019
>
> [4] Naseer et.al “On Generating Transferable Targeted Perturbations”, ICCV, 2021

---

### Official Review · Reviewer_TDqC · 2022-10-25

**Confidence:** 3
**Correctness:** 2
**Technical Novelty And Significance:** 3
**Empirical Novelty And Significance:** 3
**Recommendation:** 6

**Clarity, Quality, Novelty And Reproducibility:**

Overall clarity of the writing is fine. There are a few confusions as detailed above which I expect the authors to clarify in the rebuttal period. The formulation of the solution is novel to my knowledge, and I have no particular concerns on its reproducibility.

**Strength And Weaknesses:**

## Pros
- Incorporating temporal prompts appears to be a promising strategy as it significantly improves the transferability of attacks compared to using image models alone, when evaluated on various datasets and models.
- The proposed method also seems to have a small computational overhead as it involves learning a single attention layer while preserving the weights of the pretrained image model.
- A variety of visualizations are provided for understanding the generated adversarial patterns.

## Cons
- I am not sure I fully understand process of "mimicking dynamic cues on static images" (figure 2). Can the authors explain how the dynamic cues are created by varying the spatial scale? What would be the input to the prompting module $\cal T$ in this case? I am also confused about its role in the proposed system, as the prompting module is trained on video data (bottom of page 6). I assume mimicking is only used when evaluating on image datasets, but would appreciate if the authors can provide further clarifications.
- While the prompting module enjoys the benefit of efficiency, I am not sure if a single attention layer is sufficient to capture complex temporal dynamics. Table 5 confirms that spatial cues are retained in the temporal prompt learning (which is expected as the backbone encoder is frozen), but the accuracy on temporal-heavy dataset SSv2 is still quite low, even compared to small spatiotemporal networks. I think additional experiments are needed to show that models trained with prompting indeed learn meaningful temporal cues (e.g. testing with original vs. shuffled frame ordering), instead of relying merely on spatial features. I also think more work is needed to understand if the improvement in adversarial transfer can entirely be attributed to temporal cues, since the difference introduced by temporal prompting is larger on UCF and HMDB, which have less temporal cues than SSv2.
- Alternative ways have been studied to adapt pretrained image models to video inputs (e.g. I3D, TimeSformer). While they might not be as efficient as temporal prompting, it would still be an important comparison between proposed method and using those video models as surrogate in terms of adversarial transferability.
- There has been prior work on adapting adversarial samples between image and video modalities (e.g. https://arxiv.org/abs/2112.05379). There should be discussion/comparison to these related works.
- Results on transfer from ensemble models can be better presented. In figure 4 it is difficult to compare across rows of bar plots, or to single-model results from the main tables.

**Summary Of The Paper:**

his work studies the role of temporal dynamics in adversarial transfer between image and video models. The authors introduce temporal prompts to adapt pretrained image models to video inputs, giving them the capacity to capture dynamic cues. Motivated by visual prompt tuning, patches from video clips are processed by a self-attention layer and spatially pooled into one prompt token per frame. Spatial tokens from a single video frame and the temporal prompts are then concatenated and fed to the pretrained image classifier. Experiments show that when temporal prompts are used in surrogate model, transferred adversarial examples have a higher success rate on various target video models.

**Summary Of The Review:**

The paper presents an interesting approach towards adversarial transfer from pretrained image models to video tasks. While the proposed method has its own strengths and shows decent performances compared to baselines, I have a few concerns with the validity of hypothesis made in the paper on temporal modeling, as well as the absence of certain comparisons. Therefore, I set my initial rating as borderline reject and look forward to responses and clarifications from the authors over the discussion period.

Update 12/12: Increased rating after rebuttal and discussions with authors. See details in thread.

---

> ### Author Response · Authors · 2022-11-19
> **Response to Reviewer TDqC (1/3)**
>
> We thank the reviewer for the encouraging and insightful comments. Please find our responses to specific queries below.
>
> **Clarification on "mimicking dynamic cues on static images”.**
>
> Please note that in the case of images, where no temporal information is available, we mimic dynamic cues on scale space. We train our transformation on small scale images (Section 3.2.1). We train these transformations at multiple scales. Therefore, our adapted models are optimal across changes in scale space for image datasets such as ImageNet. The benefit of our approach is presented in Fig. 2. For example, Deit-T performance increases significantly from 7.7% to 45% on ImageNet validation set at 96x96 resolution. However, during the attack, the input to these adapted models, pre-trained on small scales, is 224x224. This leads to non-trivial gains  in adversarial transferability (**Table 3 in our paper**).
>
> **Better modeling of the temporal cues.**
> It’s a common practice to increase the number of frames  for better temporal modeling. In a similar spirit, we also increase number of training frames and consequently the number of our temporal prompts through transformation. As expected, this strategy leads to better performance in modeling dynamic cues and attack transferability (**please see Table 11 in Appendix G**).
>
> The following table shows the effect of the number of our temporal prompts on adversarial transferability.
> |  | | |  | |  |  | | |  |
> | -- | :-: | :-: | :-: | :-: | :-: | :-: | :-: | :-: | :-: |
> |        |                  | TimesFormer | TimesFormer | TimesFormer | TimesFormer | ResNet3D | ResNet3D | ResNet3D | ResNet3D |
> | Model  | Temporal Prompts | UCF         | HMDB        | K400        | SSv2        | UCF      | HMDB     | K400     | SSv2     |
> | Deit-B |       8           |       55.5      |      12.6       |      40.6        |    16.8        |     37.3     |      11.4    |     21.5     |    13.3      |
> |        |             16     |     51.4        |      10.6       |        38.7     |       15.3      |     35.2     |     9.9     |       20.6   |      15.5    |
> |        |             32     |      **21.5**       |       **6.5**       |         **23.1**     |      **10.4**       |      **19.6**     |   **6.0**        |       **13.9**    |      **11.2**     |
> |        |                  |             |             |            |             |          |          |          |          |
> | DINO   |         8         |      45.2       |    7.3          |       35.8      |      15.8       |     32.2     |    10.6      |     18.3     |  12.0        |
> |        |            16      |       42.9      |      6.0     |        35.3       |         14.5    |       29.7   |       8.8   |        18.1  |    11.5      |
> |        |              32    |       **30.2**       |        **5.7**    |        **22.6**       |          **8.3**    |      **15.3**     |       **4.9**    |        **10.0**   |       **7.3**    |
>
> _Effect of Number of Temporal Prompts on Adversarial Transferability: We observe non-trivial gains in adversarial transferability as we increase our temporal prompts. This indicates that the more the temporal information the better the attack transferability (DIM) of our approach (lower is better)._
>
>
> **Evidence of temporal cues induced by our approach.**
>
> * As suggested by the reviewer, we provide inference results on shuffled video frames in **Table 10 in Appendix F**. Our models are robust to video frame shuffling.
>
> | | | |  | | |
> | --| :-: | :-:  |:-:  |:-:  |:-:  |
> | Model | Temporal Prompts | UCF | HMDB  | K400 | SSv2|
> | Deit-B     | ☑       | 80.31$\pm$ 0.33    |  46.2$\pm$0.22 | 57.1$\pm$0.35| 16.9$\pm$0.52|
> | DINO    | ☑         |    78.40$\pm$0.35  |46.1$\pm$0.19 |55.7$\pm$0.21|17.1$\pm$0.32|
> | CLIP    | ☑         |   85.6 $\pm$ 0.23  | 55.0 $\pm$ 0.18 | 66.2 $\pm$ 0.33| 19.1 $\pm$ 0.21|
>
> _Robustness to Temporal Variations: We repeat inference results three times on a single view with 8  randomly sampled and shuffled frames. Our adapted image models show higher robustness to such temporal variations._
>
> * We show that our method improves by increasing the number of temporal tokens **Table 11 in Appendix G**. This indicates that the more the temporal information the better the attack transferability of our approach.
>
> * Our attack performs well on temporally rich datasets such as K400 (Tables 1 and 2 in our paper). We observe that our attack performance improves on SSv2 as well by increasing the number of temporal prompts (**Table 11 in Appendix G**).
>
> * Attention roll-out [1]  shows that our learned temporal prompts pay more attention to scene dynamics over time in a video as compared to original image models (**Fig. 6 in Appendix E**).

---

> > ### Author Response · Authors · 2022-11-19
> > **Response to Reviewer TDqC (2/3)**
> >
> > **Comparison against special networks designed for videos.**
> >
> > Adversarial transferability between surrogate and black-box models which share some design similarity can be higher e.g. from one Resnet model to another or its closest relative such as DesneNet. This means that the success rate of an attack can be higher if the attacker can approximate the design of the black-box model. Similarly, we can expect higher transferability from video-to-video models if the surrogate and the black-box model share architectural similarities.
> >
> > In this work, we motivate the adversarial transferability from relatively simpler image to videos models. Our approach allows launching attacks from image models to other data modalities without redesigning the specialized surrogate model for each data modality and task. Since, this works mainly considers adversarial transferability from image to video models, therefore we consider image models without temporal cues as our baseline.
> >
> > However, as suggested by the reviewer, we also compare our approach by using pure video models as surrogates; Timesformer [2], ResNet3D [3], I3D [4]. We observe that our adapted image models based on DINO compare favorably well against video surrogates and show state-of-the-art transferability (**please see Tables 6 and 8 in Appendices B and C**). This is attributed to the fact that our attack takes into account the two different representations (image and video) from the same image model.
> >
> > The following table shows a comparison of adversarial transferability between our adapted image and pure video models
> >
> > |               |             |                  |  |   | |  |    |   |
> > | -- | -- | :-: | :-: | :-: | :-: | :-: | :-: | :-: |
> > |               |             |                  | UCF|UCF|UCF| HMDB | HMDB   |HMDB   |
> > | Source Domain | Model       | Temporal Prompts | I3D  | TimesFormer | ResNet3D | I3D  | TimesFormer | ResNet3D |
> > | K400          | TimesFormer | --               | 35.0 | --          | 42.1     | 21.3 | --          | 21.1     |
> > |               | I3D         | --               | __   | 75.6        | 47.9     | --   | 25.1        | 19.0     |
> > |               | DINO (Ours) | &#9745;          | **13.2** | **63.2**        | **41.5**     | **8.4**  | **22.9**        | **15.8**     |
> > |               |             |                  |  |   | |  |    |   |
> > | SSv2          | TimesFormer | --               | 38.2 | --          | **40.6**     | 22.6 | --          | 21.0     |
> > |               | I3D         | --               | __   | 78.0        | 45.0     | --   | 29.1        | 18.9     |
> > |               | DINO (Ours) | &#9745;          | **18.7** | **62.9**        | 42.6     | **10.9** | **21.6**        | **16.1**     |
> >
> > _Cross-Task attack with Temporal Prompts: Adversarial attack using the self-supervised image model DINO with our dynamic cues (Section 2.2) performs favorably as compared to pure video models._
> >
> > The following table shows the performance of pure video models.
> > | | | |  |
> > | --| :-: | :-:  |:-:  |
> > | Dataset | TimesFormer | ResNet3D | I3D  |
> > | UCF     | 90.6        | 79.9     | 79.7 |
> > | HMDB    | 59.1        | 45.4     | 43.8 |
> > | K400    | 75.6        | 60.5     | 73.5 |
> > | SSv2    | 45.1        | 26.7     | 48.4 |
> >
> > _Generalization of Video Models: The top-1 (% ) accuracies on the validation datasets are reported. We provide single view inference on 8 frames to be consistent across these video models._
> >
> > **Differences from [6]**
> >
> > Please note [6] considers videos as a set of separate images/frames, which they forwardpass through the pre-trained image model one by one and minimize the similarity between clean and adversarial frames.
> >
> > * [6] does not introduce any dynamic cues within pre-trained image models. Therefore image model lacks the ability to exploit temporal information during attack.
> > * Frame-by-frame inference increase the computational cost as compared to our method (please also see response to reviewer iNv7 “Full-fine tuning vs. Prompt tuning and attack fully fine tuned model”)
> >
> > Kindly note that attack proposed in [6] is exactly our baseline in Tables 1 and 2 (“without temporal prompts”). Thank you for bringing our attention to [6]. We cite this work in our paper.

---

> > > ### Author Response · Authors · 2022-11-19
> > > **Response to Reviewer TDqC (3/3)**
> > >
> > > **Figure 4.**
> > >
> > > We redraw Figure 4 and compare the ensemble results with the best-performing single model within that ensemble. Specifically, we compare different network ensemble {Deit-T, Seti-S, Deit-B} with Deit-B. Different training ensemble {Deit-B, DINO, CLIP} is compared against DINO. We observe that ensemble of different training performs poorly as compared to single model DINO mainly because of CLIP textual biases within adversarial perturbations (**please see Appendix D**). Please also see the response below to “Analyses on CLIP’s low adversarial transferability”.
> > >
> > > The following table presents an analysis of textual bias within vision encoders studied in our work.
> > > | | | |
> > > | -- | :-: | :-:  |
> > > | Vision Encoder | Text Encoder | Cosine Similarity |
> > > | CLIP     | CLIP      | 0.31     |
> > > | Deit-B   | CLIP        | 0.005     |
> > > | DINO    | CLIP        | 0.002     |
> > >
> > > _Textual Bias of Vision Encoders: Results are reported on 1000 randomly selected images (one per class) from ImageNet validation set. We measure the similarity between the refined class token output by vision encoders with textual features output by CLIP text encoder for text label input as ``a photo of a \{class name\}"._
> > >
> > > **References**
> > >
> > > [1] Samira Abnar et. al “Quantifying Attention Flow in Transformers”, ACL 2020
> > >
> > > [2] Gedas Bertasius “Is Space-Time Attention All You Need for Video Understanding?”, ICML, 2021
> > >
> > > [3] Du Tran “A closer look at spatiotemporal convolutions for action recognition”, CVPR, 2018
> > >
> > > [4] Joao Carreria “Quo Vadis, Action Recognition? A New Model and the Kinetics Dataset”, CVPR, 2017
> > >
> > > [5] Naseer et.al “On Generating Transferable Targeted Perturbations”, ICCV, 2021
> > >
> > > [6] Wei et.al “Cross-Modal Transferable Adversarial Attacks from Images to Videos”, CVPR, 2022

---

> > > > ### Comment · Reviewer_TDqC · 2022-11-23
> > > > **Response to rebuttal**
> > > >
> > > > I appreciate the detailed response and additional results by the authors.
> > > >
> > > > - **Dynamic mimicking**: I think I understand the approach better now—the image transformer & prompting module are trained with  images with varying resolution from 56px to 224px, but uses full resolution during the attack. There are still some confusions on the implementation though:
> > > >   - It is unclear how the model is trained with multiple resolutions. Does the random sampler in fig. 1 select one scale from (56, 96, 120, 224) for each training image, or do the author train a separate model for each scale and use their ensemble? Does the transformation T use a single image or images of multiple scales to generate the prompts? Furthermore, is temporal prompting still needed/beneficial when training with a single image?
> > > >   - I am also not sure of the point to highlight the performance at different scales vs. DeiT, since 1) DeiT is put at disadvantage by not being fine-tuned at lower resolutions; 2) higher accuracy at lower scale does not necessarily translate to high adversarial transferability at high resolution. I think it might be a better idea to replace fig. 2 with block diagram/algorithm that details the training procedure for image models.
> > > >   - It can be misleading to present the approach as "mimicking temporal dynamics", which suggests that the scale changes over time, not just training on images of multiple scales.
> > > >
> > > > - **Temporal modeling**: I don't think robustness to clip shuffling is a positive sign for temporal modeling—a *temporal* model exploits the dynamics from frame ordering, while a *spatial* model is unaffected by shuffling the frames.
> > > >   - On SSv2, for example, frame ordering is important for discriminating classes like "Covering something with something" and "Uncovering something". The results suggest that the proposed image model relies primarily on spatial cues, which explains its lower effectiveness on temporal-heavy datasets.
> > > >   - This is not to say the proposed method cannot benefit from larger number of frames, but I suspect it does not use the dynamic cues as claimed in the paper, and instead performs similar to an ensemble of image classifiers for individual frames.
> > > >
> > > > - **Ensemble models**: Thanks for making the diagrams much more comprehensible. The ensemble of architectures appears effective, though somewhat marginal given the extra compute cost. Since CLIP is underperforming other training schemes, it would be nice to evaluate the ensemble of DeiT+DINO instead.
> > > >
> > > > Overall, despite the strong experimental results and insights to the adversarial transferability of different pretrained models, I am not confident that the paper meets the standards of publication in its current state, due to the lack of clarity and some unsupported claims. I would appreciate if the authors can further clarify the concerns above.

---

> > > > > ### Author Response · Authors · 2022-11-24
> > > > > **Response to Reviewer TDqC (1/3)**
> > > > >
> > > > > We thank the reviewer for the feedback. Please find below our responses.
> > > > >
> > > > > **Dynamic mimicking**
> > > > >
> > > > > Please note that our transformation is based on a self-attention layer followed by a pooling operation such that it generates a single prompt per frame (Section 2.1.1 “Temporal Prompts Through Transformation”). This means that our transformation generates only a single prompt if only one frame/image is available.
> > > > >
> > > > > **Do the authors train a separate model for each scale and use their ensemble?**
> > > > >
> > > > > Yes, as mentioned above “**Clarification on mimicking dynamic cues on static images**”. We train our transformations at multiple scales for static images i.e., one transformation per resolution scale. Therefore for a single image model, for example, Deit-B, we obtain multiple transformations optimal for different resolutions (56, 96, 120).  We attack the ensemble of transformations to benefit from change in scale space for the boost in adversarial transferability (Table 3 in our paper).
> > > > >
> > > > > **Does the transformation T use a single image or images of multiple scales to generate the prompts?**
> > > > >
> > > > > The input to transformation T is a single image for static image datasets. We believe that the multiple transformations learned at different scales bring diversity to adversarial perturbations and enhance transferability as indicated by our results.
> > > > >
> > > > > **Is temporal prompting still needed/beneficial when training with a single image?**
> > > > >
> > > > > We apologize for the confusion caused due to the term temporal prompting, which we adopted since our major focus was to transfer image models to videos. However, ours is a generic adaptation strategy that can be easily adapted for other transfer cases, e.g., as we show for the multiple spatial resolution case.
> > > > >
> > > > > Please note that our approach improves a pre-trained model’s ability to process images at different resolutions (Fig. 2 in our paper, please also see the table below). This diversity in representation learning at multiple scales allows optimizing for transferable perturbation as indicated by the results in our paper. The attack results without our transformations reduce significantly (please see the table below). This indicates the effectiveness of our method.
> > > > >
> > > > > **DeiT is put at disadvantage by not being fine-tuned at lower resolutions**
> > > > >
> > > > > Kindly note that image models overfit at a given training resolution and are not able to perform better at different test time low resolutions. For example, we observe performance degradation in image modes (trained at the resolution of 224) at lower resolutions as shown in the following table:
> > > > >
> > > > > | | | | |
> > > > > | -- | :-:| :-:| :-: |
> > > > > | Method                                       | **Deit-T**               | **Deit-S**               | **Deit-B**               |
> > > > > |                                              | Res: 56 — 224 | Res: 56 — 224 | Res: 56 — 224 |
> > > > > | Pre-trained on 224                           | 0.7 — 72.2           | 1.3 — 79.9           | 2.3 — 81.8           |
> > > > > | Pre-trained on 224 and Fine-tuned on 56      | 21.5 — 49.1          | 36.7 — 58.7          | 45.5 — 65.8          |
> > > > > | Ours [Pre-trained on 224 and adapted for 56] | 16.6 — 72.2          | 28.1 — 79.9          | 38.1 — 81.8          |
> > > > >
> > > > > _Our approach adapts the image models at low resolution without losing the ability to perform at higher resolutions thereby striking the right balance at different resolution scales for image datasets. We report top-1 (%) accuracy on ImageNet validation set. The first—second values represent accuracies at resolutions of 56 and 224, respectively._
> > > > >
> > > > > Please note that our aim is to adapt pre-trained image models for lower resolutions without losing their ability to perform at higher resolutions. This is similar to adapting image model to perform better on videos.  As we observe for videos,  if we simply fine-tune on lower resolution then the model simply loses the original spatial representation at higher resolution (e.g., 224).
> > > > >
> > > > > Thus, our approach enhances the performance of the image models for videos and low resolution image datasets while preserving the original image representations. These representations in turn enhance the transferability of the adversarial attack.

---

> > > > > > ### Author Response · Authors · 2022-11-24
> > > > > > **Response to Reviewer TDqC (2/3)**
> > > > > >
> > > > > > **Higher accuracy at lower scale does not necessarily translate to high adversarial transferability**
> > > > > >
> > > > > > Kindly note that low resolution network training followed by high resolution fine-tuning does provide a regularization effect for clean images [4]. Similarly, low resolution adaption followed by high resolution adversarial attack provides a regularization effect to adversarial perturbations. This is also indicated by our results as the adversarial transferability of the original image models (“without our learned transformations at varying scale space”) is significantly lower as shown in the following table:
> > > > > >
> > > > > > | | |||  | | |  |  |  |  |  | |
> > > > > > | :-: | :-:| :-: | :-: | :-: | :-:| :-:| :-: | :-: | :-:| :-: | :-:| :-: |
> > > > > > | Model  | Method           | No. of Ens. | Type of Ens.   | Params (M) | BiT50 | Res152 | WRN   | DN201 | ViT-L | T2T24 | TnT   | T2T7  |
> > > > > > | Deit-B | --               | 0                   | --                  | 86                    | 56.24 | 59.14  | 60.64 | 64.44 | 61.38 | 69.54 | 73.96 | 64.44 |
> > > > > > | Deit-B |  [3] | 12                  | Self-ensemble       | 157                   | 80.10 | 84.92  | 86.36 | 89.24 | 78.90 | 84.00 | 92.28 | 93.42 |
> > > > > > | Deit-B | Ours             | 3                   | Resolution ensemble | 121                   | **86.64** | **90.88**  | **92.14** | **93.82** | **95.64** | **95.74** | **98.48** | **94.74** |
> > > > > >
> > > > > > _Fool rate (%) of DIM attack on 5k images from ImageNet validation set [3].  Our approach has state-of-the-art transferability rates with lower parameter (in Millions) complexity._
> > > > > >
> > > > > > Our ensemble of different transformations learned at only three low resolution scales (56, 96, and 120) increases transferability in comparison to [3] which uses a self-ensemble of 12 refinement blocks.
> > > > > >
> > > > > >
> > > > > > **Presenting the approach as “mimicking temporal dynamics", which suggests that the scale changes over time, not just training on images of multiple scales.**
> > > > > >
> > > > > > Kindly note that we refer to our approach for static images as “mimicking change across scale-space” (Section 2.1.1). As recommended, we will further highlight and clarify the difference in dynamic behavior modeled via our approach for static images (change in scale-space) and videos (change in temporal information).
> > > > > >
> > > > > > **Temporal Modeling**
> > > > > >
> > > > > > **On SSv2, for example, frame ordering is important for discriminating classes like "Covering something with something" and "Uncovering something". The results suggest that the proposed image model relies primarily on spatial cues, which explains its lower effectiveness on temporal-heavy datasets.**
> > > > > >
> > > > > > Kindly note that existing video models which are dedicatedly designed to model temporal behavior also retain strong performance under frame shuffle. To illustrate this,  we apply the same experiment of frame shuffling (suggested by the reviewer) to TimesFormer [1] and ResNet3D [2] for K400 as well as SSv2 dataset.  We do not observe significant performance degradation against frame shuffling operation for these well-performing video models. The results are in the table below.
> > > > > >
> > > > > > |                 |           |                  |                 |
> > > > > > | :-: | :-:  | :-: | :-:  |
> > > > > > | Model           | Shuffling | K400             | SSv2            |
> > > > > > | TimesFormer [1] | x         | 75.6             | 45.1            |
> > > > > > | TimesFormer [1] | &#9745;   | 75.8 $\pm$ 0.1 | 43.7 $\pm$4.5 |
> > > > > > | ResNet3D [2]    | x         | 60.5             | 26.7            |
> > > > > > | ResNet3D [2]    | &#9745;   | 57.9 $\pm$1.3  | 26.3 $\pm$5.9 |
> > > > > >
> > > > > > _Robustness to Temporal Variations: We repeat inference results three times on a single view with 8 randomly sampled and shuffled frames. We observe that pure video models trained for temporal modeling retain performance to frame shuffling operation, a phenomenon similar to our adapted image models for videos._
> > > > > >
> > > > > > These results show that the random shuffle may not be ideally suited to check the suitability of temporal trends learned by the model since random shuffling may still retain enough structure to predict the correct class. To clearly study the behaviour that the reviewer is trying to check, we test on SSv2 videos for “something covering something” and “uncovering something” by totally reversing each sequence and then evaluating our model. We visualize results for several qualitative examples and note that our model indeed confuses the totally reverse sequence as another action class. The examples are provided at the following anonymous link: https://anonymous.4open.science/r/ICLR_Rebuttal-1982/README.md.
> > > > > >
> > > > > > We thank the reviewer and will include these observations in appendix F “Analysing Dynamic Cues Against Temporal Variations” in our final draft.

---

> > > > > > > ### Author Response · Authors · 2022-11-24
> > > > > > > **Response to Reviewer TDqC (3/3)**
> > > > > > >
> > > > > > > **Ensemble of Self-supervised (DINO) and Supervised (Deit) Models**
> > > > > > >
> > > > > > > Yes, as expected, adversarial transferability from an ensemble of our adapted DINO and Deit increases than from any of the individual models (DINO, Deit or CLIP). As recommended, we will add these results as well in our final draft.
> > > > > > >
> > > > > > > |  |  |  | |  |  |  |  | | |
> > > > > > > | -- | :-: | :-: | :-:| :-: | :-: | :-: | :-: | :-: | :-: |
> > > > > > > |  | | | | TimesFormer | | | | ResNet3D ||
> > > > > > > | Model          |Temporal Prompts | UCF         | HMDB        | K400        | SSv2        | UCF      | HMDB     | K400     | SSv2     |
> > > > > > > | Deit-B         | &#9745;          | 55.5        | 12.6        | 40.6        | 16.8        | 37.3     | 11.4     | 21.5     | 13.3     |
> > > > > > > | DINO           | &#9745;          | 45.2        | 7.3         | 35.8        | 15.8        | 32.2     | 10.6     | 18.3     | 12.0     |
> > > > > > > | {Deit-B, DINO} | &#9745;          | **39.1**        | **5.0**         | **31.7**        | **12.9**        | **28.5**     | **8.7**      | **15.9**     | **10.1**     |
> > > > > > >
> > > > > > > _The ensemble of supervised and self-supervised image models with temporal cues,  Deit and DINO, further improves adversarial transferability to video models. Results (top-1 %) are reported for the DIM attack._
> > > > > > >
> > > > > > > We hope these additional experimental evidence clarifies our claims about temporal modeling for videos as well as for applying our approach to static image-only datasets.
> > > > > > >
> > > > > > > We appreciate and value your feedback. Any further discussion/questions are welcome! Your support of our novel approach that identifies new insights and mechanisms about adversarial transferability is very important and we sincerely appreciate it.
> > > > > > >
> > > > > > > **References**
> > > > > > >
> > > > > > > [1] Gedas Bertasius “Is Space-Time Attention All You Need for Video Understanding?”, ICML, 2021
> > > > > > >
> > > > > > > [2] Du Tran “A closer look at spatiotemporal convolutions for action recognition”, CVPR, 2018
> > > > > > >
> > > > > > > [3] Naseer et.al  “On Improving Adversarial Transferability of Vision Transformers ”, ICLR,2022
> > > > > > >
> > > > > > > [4]Touvron et.al “Deit iii: Revenge of the vit”, ECCV,2022

---

> > > > > > > > ### Comment · Reviewer_TDqC · 2022-12-12
> > > > > > > > **Thank You**
> > > > > > > >
> > > > > > > > I would like to thank the authors again for the follow-up discussions, and will update my rating based on the additional results.
> > > > > > > >
> > > > > > > > - Dynamic mimicking: I appreciate the clarifications, though most of these important details are not apparent from the paper. It is not clear from the current submission how a temporal prompting module designed for videos can be easily adapted to image inputs by varying resolution, and why it should be effective under such a different domain. To me the connection between the "scale-space" of images and the "temporal cues" of videos is not strong enough, making the section on dynamic mimicking somewhat detached from the rest of paper.
> > > > > > > >
> > > > > > > > - Temporal dynamics: While I understand that frame shuffling is not a perfect way to evaluate temporal modeling, some architectures are more sensitive to temporal cues than others (e.g. [MViT](https://arxiv.org/abs/2104.11227), table 9). I am not yet convinced that temporal cues are the main contributor to the improved adversarial transferability, given that 1) the adapted models report low accuracies on all video datasets, especially SSv2 (~20%, compared to video SOTA of 60+%); and 2) the prompting module shows similar improvements on images of varying resolution.
> > > > > > > >
> > > > > > > > Given the above, I feel that it might be better to de-emphasize the temporal component in the paper, and instead focus on how the prompting module can increase the robustness of adapted transformers, subsequently improving adversarial transfer across video and image datasets.

---

> > > > > > > > > ### Author Response · Authors · 2022-12-13
> > > > > > > > > **Thank you for the concluding remarks.**
> > > > > > > > >
> > > > > > > > > We thank the reviewer for the concluding remarks. We value your suggestion and will update the final draft accordingly.

---

### Official Review · Reviewer_iNv7 · 2022-10-25

**Confidence:** 3
**Correctness:** 2
**Technical Novelty And Significance:** 3
**Empirical Novelty And Significance:** 3
**Recommendation:** 5

**Clarity, Quality, Novelty And Reproducibility:**

The problem settings and some parts of the proposed method are not clearly stated to some extent. The problem is new and there are some novel ideas. The reproducibility may depend on the release of code.

**Strength And Weaknesses:**

[Strength]
1. The problem studied in this paper, i.e., using image models to attack black-box video models, is interesting and important in practice, as there are a large and increasing number of public pre-trained image models.
2. The proposed method is simple and seems effective. The idea of mimicking dynamic cues on static images with different spatial scales is interesting, which can improve the adversarial transferability on image data.

[Weakness]
1. The problem settings are not clearly stated. For example, it can only be inferred from the method and experiments that the attacker has the information on which dataset the target model is trained on and also the access to this dataset.
2. Some of the claims are not well supported.
- Abstract: "Our attack results indicate that the attacker does not need specialized architectures ... Image models are all we need
to optimize for an effective surrogate ..." Without baseline results on surrogate models with those "specialized architectures", it cannot be determined whether the proposed method is really "effective" compared with the baselines.
- Section 4.1 (Table 5): "Deit models retain their top-1 (%) accuracy on ImageNet ..., while also exhibiting decent performance on video datasets and ..." To support this claim, it is expected that Table 5 should contain: (1) the ImageNet results for raw Deit models; (2) a baseline result for video datasets.
- Section 4.1.1 (Figure 4): "Ensemble of different training frameworks performs favorably to boost attack transferability." Visually comparing the two rows in Figure 4 to figure out which is better can be difficult. A better demonstration is preferred.
3. In the proposed method shown in Figure 1, a random frame is sampled from the input video. It is not stated whether the input receives gradient from this branch during attacks. Besides, does this single frame sampling strategy limits the attack to the video data where any single frame contains a large portion of information of the video?
4. It is discovered that "CLIP adversaries are relatively less transferable as compared to fully-supervised ViT or self-supervised DINO model". This may be weird since CLIP is pre-trained on a wider range of data. Is there any reasonable explanation?
5. The paper still learns the downstream task when attacking the video task, which only replaces fine-tuning with visual prompt tuning. This adversarial attack transfer is not a direct attack on the image model. Considering Transformation and the image model as a whole, it can be considered as a video model learned by the prompt tuning method.  And although visual prompt tuning needs to learn fewer parameters, the training time is not shorter than fine-tuning.
6. The prompt tuning approach usually does not perform as well as the fine-tuning approach. Would an attack using a fine-tuning model on a downstream video task perform better?
7. The paper only compares the use of Temporal Prompts in the experimental section. Currently, some methods have been proposed to improve the cross-task or cross-model transferability of adversarial attacks, and these methods should also be involved in the comparison.
8. The approach of the paper only tries to attack the transformer model and has limited transferability to CNN models. Can this approach also be used to attack pre-trained CNN models, such as MOCO for Resnet?


**Summary Of The Paper:**

This paper proposes a transfer-based adversarial attack for black-box video models. The surrogate model is built via prompt tuning from a pre-trained image model with additional temporal prompt tokens. It is validated that a surrogate model with the learned temporal prompts (or "dynamic cues") achieves better adversarial transferability for various attacks than a surrogate model without them. This approach can also extend to target models on image or 3D data.

**Summary Of The Review:**

The problem studied in this paper is meaningful and the proposed method is intuitive. The major issue is the unsupported claims with some important baselines missing. Besides, the writing and organization can be improved to make it easier to understand the method and settings.

---

> ### Author Response · Authors · 2022-11-18
> **Response to Reviewer  iNv7 (1/4)**
>
> We thank the reviewer for the encouraging and insightful comments. Please find our responses to specific queries below.
>
> **Regarding the problem setting.**
>
> Kindly note that we followed standard adversarial transferability protocol [1,2]; the attacker is unaware of the black-box video model but has access to its train data. Our temporal cues also boost cross-task adversarial transferability with no access to the black-box video model, its training data, or label space ( **please see Appendix B and Table 6**). As recommended, we describe these settings in the experimental section of our paper (Section 4, highlighted in red).
>
> **Baseline results and claims.**
>
> Adversarial transferability between surrogate and black-box models which share some design similarity can be higher e.g. from one ResNet model to another or its closest relative such as DesneNet. This means that the success rate of an attack can be higher if the attacker can approximate the design of the black-box model. Similarly, we can expect higher transferability from video-to-video models if the surrogate and the black-box model share architectural similarities.
>
> In this work, we motivate the adversarial transferability from relatively simpler image to videos models. Our approach allows launching attacks from image models to other data modalities without redesigning the specialized surrogate model for each data modality and task. Since, this works mainly considers adversarial transferability from image to video models, therefore we consider image models without temporal cues as our baseline.
>
> However, as suggested by the reviewer, we also compare our approach by using pure video models as surrogates; Timesformer [4], ResNet3D [5], I3D [6]. We observe that our adapted image models based on DINO compare favorably well against video surrogates and show state-of-the-art transferability (**please see Tables 6 and 8 in Appendices B and C**). This is attributed to the fact that our attack takes into account the two different representations (image and video) from the same image model.
>
> The following table shows a comparison of adversarial transferability between our adapted image and pure video models
> |               |             |                  |  |   | |  |    |   |
> | -- | -- | :-: | :-: | :-: | :-: | :-: | :-: | :-: |
> |               |             |                  | UCF|UCF|UCF| HMDB | HMDB   |HMDB   |
> | Source Domain | Model       | Temporal Prompts | I3D  | TimesFormer | ResNet3D | I3D  | TimesFormer | ResNet3D |
> | K400          | TimesFormer | --               | 35.0 | --          | 42.1     | 21.3 | --          | 21.1     |
> |               | I3D         | --               | __   | 75.6        | 47.9     | --   | 25.1        | 19.0     |
> |               | DINO (Ours) | &#9745;          | **13.2** | **63.2**        | **41.5**     | **8.4**  | **22.9**        | **15.8**     |
> |               |             |                  |  |   | |  |    |   |
> | SSv2          | TimesFormer | --               | 38.2 | --          | **40.6**     | 22.6 | --          | 21.0     |
> |               | I3D         | --               | __   | 78.0        | 45.0     | --   | 29.1        | 18.9     |
> |               | DINO (Ours) | &#9745;          | **18.7** | **62.9**        | 42.6     | **10.9** | **21.6**        | **16.1**     |
>
> _Cross-Task attack [7] with Temporal Prompts: Adversarial attack using the self-supervised image model DINO with our dynamic cues (Section 2.2) performs favorably as compared to pure video models._
>
> **ImageNet results for Deit models.**
>
> As suggested, we updated Table 5 with ImageNet results. We did not observe any major drop in ImageNet solution after training for our temporal cues. The only noticeable drop is observed within Deit-S model whose top-1 (%) on ImageNet validation decreases from 79.9% to 78.8% while adapting depth modality for ModelNet40.
>
> **Baseline results for video datasets**
>
> As mentioned above, we consider Timesformer [4], ResNet3D [5], and I3D [6] as our baselines. We report their generalization on video datasets in **Table 7 in Appendix C**. Our approach enables the attacker to use both spatial and temporal representations from image models and therefore performs better than the video models.
>
> The following table shows the performance of video models.
> | | | |  |
> | --| :-: | :-:  |:-:  |
> | Dataset | TimesFormer | ResNet3D | I3D  |
> | UCF     | 90.6        | 79.9     | 79.7 |
> | HMDB    | 59.1        | 45.4     | 43.8 |
> | K400    | 75.6        | 60.5     | 73.5 |
> | SSv2    | 45.1        | 26.7     | 48.4 |
>
> _Generalization of Video Models: The top-1 (% ) accuracies on the validation datasets are reported. We provide single view inference on 8 frames to be consistent across these video models._

---

> > ### Author Response · Authors · 2022-11-19
> > **Response to Reviewer  iNv7 (2/4)**
> >
> > **Figure 4.**
> >
> > We redraw Figure 4 and compare the ensemble results with the best-performing single model within that ensemble. Specifically, we compare different network ensemble {Deit-T, Seti-S, Deit-B} with Deit-B. Different training ensemble {Deit-B, DINO, CLIP} is compared against DINO. We observe that ensemble of different training performs poorly as compared to single model DINO mainly because of CLIP textual biases within adversarial perturbations (**please see Appendix D**). Please also see the response below to “Analyses on CLIP’s low adversarial transferability”.
> >
> > **Input gradients during attack.**
> >
> > We apply the existing attacks to our models. This means that we simply aggregate gradients from both branches to the input video frames. We clarify this in the experimental Section 4 of our paper.
> >
> > **Single frame strategy and attack performance**
> >
> > Our attack is equally effective across datasets where temporal information spreads across frames. For example, the relative decrease of Timesformer [4] on K400 is 52%  compared to UCF (50%), where objects are mostly centered across frames (DIM attack, Table 2 in our paper). This is because our transformation allows the gradient information to flow across frames. Our attack performance increases with the number of frames which indicates our attack scales well with higher temporal information (**Table 11 in Appendix G**). The transferability of DIM attack from our DINO model to TimesFormer increases by 58% as we increase the number of temporal prompts from 8 to 32. A similar trend is observed for Deit-B  as well (**Table 11 in Appendix G**). We also show results without temporal prompts which degrades the tranferability performance (Tables 1 and 2 in our paper). This provides an additional evidence for the benefit of dynamic cues.
> >
> > The following table shows the effect of the number of temporal prompts on adversarial transferability.
> >
> > |  | | |  | |  |  | | |  |
> > | -- | :-: | :-: | :-: | :-: | :-: | :-: | :-: | :-: | :-: |
> > |        |                  | TimesFormer | TimesFormer | TimesFormer | TimesFormer | ResNet3D | ResNet3D | ResNet3D | ResNet3D |
> > | Model  | Temporal Prompts | UCF         | HMDB        | K400        | SSv2        | UCF      | HMDB     | K400     | SSv2     |
> > | Deit-B |       8           |       55.5      |      12.6       |      40.6        |    16.8        |     37.3     |      11.4    |     21.5     |    13.3      |
> > |        |             16     |     51.4        |      10.6       |        38.7     |       15.3      |     35.2     |     9.9     |       20.6   |      15.5    |
> > |        |             32     |      **21.5**       |       **6.5**       |         **23.1**     |      **10.4**       |      **19.6**     |   **6.0**        |       **13.9**    |      **11.2**     |
> > |        |                  |             |             |            |             |          |          |          |          |
> > | DINO   |         8         |      45.2       |    7.3          |       35.8      |      15.8       |     32.2     |    10.6      |     18.3     |  12.0        |
> > |        |            16      |       42.9      |      6.0     |        35.3       |         14.5    |       29.7   |       8.8   |        18.1  |    11.5      |
> > |        |              32    |       **30.2**       |        **5.7**    |        **22.6**       |          **8.3**    |      **15.3**     |       **4.9**    |        **10.0**   |       **7.3**    |
> >
> > _Effect of Number of Temporal Prompts on Adversarial Transferability: We observe non-trivial gains in adversarial transferability as we increase our temporal prompts. This indicates that the more the temporal information the better the attack transferability (DIM) of our approach (lower is better)._
> >
> > **Analyses on CLIP’s low adversarial transferability**
> >
> > Our results indicate that adversarial perturbations found via the CLIP vision encoder show lower transferability as compared to similar models (in terms of network design) such as DINO [2] and Deit-B [1]. We further inspect this behavior in Appendix C (**please see Table 9 and Fig. 5**). We observe that CLIP vision features are aligned towards textual features and as a result adversaries found within CLIP models are textually biased. We observe textual patterns emerging within adversarial perturbations (**Fig. 5 in Appendix D**). These perturbations are optimal to fool CLIP but not that meaningful for vision models trained without text supervision. We quantify this as well by measuring cosine similarly between visual features of CLIP vision encoder, DINO, and Deit-B with CLIP textual features produced for text labels (**Table 9 in Appendix D**). These observations further explain the lower performance of the ensemble of different training frameworks as compared to an ensemble of different networks in Fig. 4 in our paper.

---

> > > ### Author Response · Authors · 2022-11-19
> > > **Response to Reviewer  iNv7 (3/4)**
> > >
> > > The following table presents an analysis of textual bias within vision encoders studied in our work.
> > >
> > > | | | |
> > > | -- | :-: | :-:  |
> > > | Vision Encoder | Text Encoder | Cosine Similarity |
> > > | CLIP     | CLIP      | 0.31     |
> > > | Deit-B   | CLIP        | 0.005     |
> > > | DINO    | CLIP        | 0.002     |
> > >
> > > _Textual Bias of Vision Encoders: Results are reported on 1000 randomly selected images (one per class) from ImageNet validation set. We measure the similarity between the refined class token output by vision encoders with textual features output by CLIP text encoder for text label input as ``a photo of a \{class name\}"._
> > >
> > > **Full-fine tuning vs. Prompt tuning and attack fully fine tuned model**
> > >
> > > We aim to use the same image backbone for different video or multi-view datasets. Our design has lower memory cost and computational overhead as we don't need to update all model parameters during the backpass (Table 13 in Appendix H).
> > >
> > > The following table shows the computational analysis of different models.
> > >
> > > | |  || |  |
> > > | --| :-: | :-: | :-: | :-: |
> > > | Model       | Temporal Prompts | Total Parameters (Millions) | Trainable Parameters ((Millions)) | GFLOPs |
> > > | Deit-B      | x                | 86.0                        | --                                | 140.6  |
> > > | Deit-B      | ☑                | 95.0                        | 7.7                               | **34.1**   |
> > > |             |                  |                             |                                   |        |
> > > | DINO        | x                | 86.0                        | --                                | 140.6  |
> > > | DINO        | ☑                | 95.0                        | 7.7                               | **34.1**   |
> > > |             |                  |                             |                                   |        |
> > > | CLIP        | x                | 85.6                        | --                                | 139.5  |
> > > | CLIP        | ☑                | 93.6                        | 7.4                               | **34.1**   |
> > > |             |                  |                             |                                   |        |
> > > | TimesFormer | --               | 121.7                       | 121.7                             | 190.0  |
> > >
> > > _Our approach has significantly less computational overhead with few learnable parameters. In comparison to video models such as TimesFormer, we need to train only 7 million parameters. Note that video models like Timesformer simultaneously forward pass all sampled frames through the model layers. The image models like Deit can only process one frame at a time so these models require multiple forward pass to process a video. In contrast, our single transformation layer process a video to generate temporal prompts which are combined with a randomly selected single frame to forward pass through our image models. As result, our approach is computationally cheaper than pure video or image models while increasing the adversarial transferability. We present the results for single view with 8 frames._
> > >
> > > Please note that our attack exploits pre-trained image representations (for example learned on ImageNet) along with dynamic cues (via temporal prompts) to fool black-box video models.  These adapted image models are not optimal for video representation learning as compared to specialized video networks such as TimesFormer.
> > >
> > > When we fully fine-tune the pre-trained image model with our design to videos, the performance of the adapted models improves, in terms of top-1 (%) accuracy, on video datasets (for example Deit-B top-1 (\%) increases from 81.4% to 85.7%). As a result of this fine-tuning, however, we lose the original image representations otherwise preserved in our approach. This means that the contribution of our self-supervised loss ($\mathcal{L}_{ss}$), designed to exploit image level representation, reduces and as a result, it affects the transferability of adversarial attacks (**please see Table 12 in Appendix H**).
> > >
> > > |        |                  |             |             |            |
> > > | -- | :-: | :-:| :-: | :-: |
> > > |        |                  |             | TimesFormer | TimesFomer |
> > > | Model  | Temporal Prompts | Fine Tuning | UCF         | HMDB       |
> > > | DINO   | ☑                | ☑           | 56.3        | 13.6       |
> > > |        | ☑                | x           | **45.2**        | **8.4**        |
> > > |        |                  |             |             |            |
> > > | Deit-B | ☑                | ☑           | 59.3        | 12.1       |
> > > |        | ☑                | x           | **55.5**        | **12.6**       |
> > >
> > > _Temporal Prompts Vs. Full Fine Tuning: Our proposed temporal Prompts perform favorably as compared to the full fine-tuning of the image model for dynamic cues. Full fine-tuning affects the pre-trained image representations which in turn lowers the contribution of self-supervised attack loss (Eq. 3) during the attack, reducing adversarial transferability._

---

> > > > ### Author Response · Authors · 2022-11-19
> > > > **Response to Reviewer  iNv7 (4/4)**
> > > >
> > > > Our prompt learning approach preserves the original image representation within the spatial class token which is used in our self-supervised loss to boost adversarial transferability. We lose the original image representation when doing full fine-tuning of our model on videos and as a result, transferability decreases (**Table 12 in Appendix H**).  Interestingly this tradeoff between full fine-tuning and  adversarial transferability is even more prominent with self-supervised DINO model as supervised fine-tuning reduces self-supervised inductive biases learned by the DINO.
> > > >
> > > > _In conclusion, attacks from pure image representations that is without temporal prompts, or from pure video representation that is without image representation show relatively lower transferability. The combination of adversarial image and video representations compliments each other  in enhancing the transferability to black-box videos and image models. Both image and video representations are well preserved in our approach with in a single image model._
> > > >
> > > > **Cross Task and Cross Model Transferability**
> > > >
> > > > Kindly note that our approach is complementary to the existing attacks. As recommended by the reviewer, we show cross-task transferability in Appendix B using [7] (**please see Table 6 in Appendix B**). Our temporal prompts provide clear benefits in cross-task problem settings.
> > > >
> > > > **Applicability to CNNs**
> > > >
> > > > The main focus of this work is to study the transferability properties of state of the art Vision Transformer models. Scalability of our approach to CNNs could be possible, e.g., considering the prompt tuning methods applied to CNNs [8]. However, this is beyond the scope of our current work.
> > > >
> > > > **Regarding Claim in Abstract**
> > > >
> > > > We rephrased the sentence in abstract from “mage models are all you need…. .” to “Image models are the effective surrogate to optimize an adversarial attack to fool black-box models in a changing environment over time.”
> > > >
> > > > **References**
> > > >
> > > > [1] Yinpeng Dong et.al “Boosting Adversarial Attacks with Momentum”, CVPR, 2018
> > > >
> > > > [2] Naseer et.al  “On Improving Adversarial Transferability of Vision Transformers ”, ICLR,2022
> > > >
> > > > [3] Naseer et.al “On Generating Transferable Targeted Perturbations”, ICCV, 2021
> > > >
> > > > [4] Gedas Bertasius “Is Space-Time Attention All You Need for Video Understanding?”, ICML, 2021
> > > >
> > > > [5] Du Tran “A closer look at spatiotemporal convolutions for action recognition”, CVPR, 2018
> > > >
> > > > [6] Joao Carreria “Quo Vadis, Action Recognition? A New Model and the Kinetics Dataset”, CVPR, 2017
> > > >
> > > > [7] Naseer et.al “Cross-Domain Transferability of Adversarial Perturbations”, NeurIPS, 2019
> > > >
> > > > [8] Jia et.al “Visual Prompt Tuning”, ECCV, 2022

---

> ### Author Response · Authors · 2022-12-13
> **We hope the reviewers' concerns are addressed. Any further questions/suggestions/concluding remarks are most welcome.**
>
> We appreciate the reviewers' valuable feedback. Detailed answers to the reviewers' queries are provided below.
>
> If there is any pending question, please let us know. We will be happy to answer as soon as possible.

---

### Official Review · Reviewer_yaNd · 2022-10-25

**Confidence:** 3
**Correctness:** 3
**Technical Novelty And Significance:** 3
**Empirical Novelty And Significance:** 3
**Recommendation:** 6

**Clarity, Quality, Novelty And Reproducibility:**

As discussed in the previous section, my main concern regarding this paper is the clarity of writing and method definition, and consequently, reproducibility. The evaluation seems thorough. I lack deep knowledge of prior work to judge its novelty.


**Strength And Weaknesses:**

Strengths:
The problem is well-motivated - it is clear why one might want to adjust the surrogate model when going from images to videos
Results are promising - the resulting model bests all baselines by a significant margin
Evaluation is thorough and shows similar trends across a wide range of datasets, surrogate models, adversarial attacks, resolutions, and ensembling techniques.
Weaknesses:
I am not familiar with prior work on adversarial transferability, so my confusion might be stemming from that. Nevertheless, I found the paper quite difficult to follow. The syntax in the introduction is sometimes difficult to parse, but is generally understandable (“in a real-world setting, a scene is not static but mostly involves various dynamics, e.g., object motion”,  “approach offers the benefit that the attacks do not need to rely on specialized networks designed for videos towards better adversarial transferability”, “dynamic cues are shown here to aid in better transfer not only to video models, but also to the original domain (image) models due to the expressivity of dynamic information”). The method section is more difficult to follow, because

1) The method is introduced only in a figure and scattered throughout two pages of plain text - no model expression summarizing how final outputs are obtained via inputs, no clear definition of used losses;
2) After re-reading the method section and following the diagram several times, I am still confused about some of the wording.

Why does the caption in Figure 1 say that “the spatial tokens are ignored”?

Do authors use the term “token” and the output of the transformer at that token’s position interchangeably? (e.g. I_cls in the first paragraph of Sec 3.1)

In sec 3.2 “spatial class token Icls serves in an unsupervised objective” - means that it is used only in the unsupervised objective? Is self-supervised L_ss - that unsupervised objective?

I also found it difficult to follow the line of thought in the experiment results section 4.1 - it is only a few dozen lines long and it basically lists all provided Tables, and follows straight to conclusions, not connecting observations to conclusions in the main text.



**Summary Of The Paper:**

Authors explore the problem of robustifying a video transformer using a pre-trained image transformer as a surrogate model. Authors argue that using adversarial patterns optimal for large image models (such as Deit, DINO and CLIP) improves adversarial robustness of other image models, but is not suitable out-of-the-box for robostifying video models because image models lack temporal cues (e.g. motion). Authors train a network that generates a sequence of temporal tokens (one per frame) for the entire video, concatenate them with spatial tokens (one per spatial location) for a single random frame, and pass through a frozen pre-trained image transformer with fine-tuned heads. The resulting pipeline is trained using a combination of supervised and unsupervised losses and is then attacked to obtain adversarial examples. Authors train their method across multiple scales to improve transferability and mimic scale changes in videos on image datasets. Authors show that in the resulting pipeline, larger image models yield better image-to-video adversarial transferability, self-supervised DINO transfer best, that their approach is more robust then 3D convolution, that the ensembling helps their approach as well. Authors claim that the resulting procedure is significantly more memory efficient compared to a naive baseline of processing all spatial tokens of all frames jointly.


**Summary Of The Review:**

I think that the paper does a good job of motivating the problem it tackles, and that results reported in this paper would benefit the community at large, but the lack of clarity severely undermines the impact this publication can make in its current form.

---

> ### Author Response · Authors · 2022-11-18
> **Response to Reviewer yaNd (1/3)**
>
> We thank the reviewer for the encouraging and insightful comments. Please find our responses to specific queries below.
>
> **Model expression (input, output) and definition of used losses?**
>
> We demonstrate our approach using three types of Vision Transformers [1,2,3]. These models have a standard design and are sequentially composed of “n” number of multi-head self-attention (MHSA) blocks [4]. Different from convolutional networks, ViTs divide an input image into several patches, known as ‘patch tokens’. A randomly initialized vector, also called a ‘class token’, is usually appended with these patch tokens. We collectively refer to these class and patch tokens as 'spatial tokens’ (Fig. 1 in our paper). The models process this set of tokens and produce an output of the same size as that of input. We refer to the model output as ‘refined tokens’. In the case of classification, the refined class token is extracted from the model output and projected by an MLP layer (spatial head in Fig. 1) to predict the class category.
>
> We adapt these models by introducing ‘temporal prompts’ (output from our transformation). In other words, we increase the input tokens into the image model by the number of temporal prompts. In this work, we set the maximum number of temporal prompts to 8. We forward pass these tokens through the model and extract the temporal prompts and the temporal class token at the model’s output. The average of these refined temporal prompts and class tokens is then projected by the temporal head (Fig. 1) to learn dynamic cues within pre-trained and frozen image models.
>
> _As recommended, we expand our section 3.1 (Preliminaries) to better define the notations for these models, inputs and outputs, learnable tokens, and the output refined tokens by the model. Consequently, we further explain forward pass for videos through our adapted image models defined in Eq. 1. Please note that our approach is complementary to existing attacks. Any existing attack loss functions designed for image models can seamlessly be extended to our approach. We demonstrate this for cross-task attack settings (defined in Section 4) by showing the effectiveness of our temporal prompts using cross-task attack [5] (**please see Table 6 in Appendix B**). As recommended, we define the attack loss terms used in our paper (Section 3.2) based on the above notations for the inputs, outputs, and model forward pass._
>
> _We hope this clarifies the reviewer’s concern regarding the clarity of model definitions._
>
> **Why does the caption in Figure 1 say that “the spatial tokens are ignored”?**
>
> As described above and detailed in Section 3.1, we collectively refer to the pre-trained class token within image models and patch tokens as ‘spatial tokens’. We adapt these image models with temporal prompts and class tokens and  only train these for video datasets with frozen image features. We concatenate our temporal tokens with the spatial tokens (Fig. 1) and forward pass all of these tokens through the model (Eq. 1 in our paper). The temporal tokens interact with these spatial tokens through self-attention layers within ViT. We extract only the refined temporal prompts at the model’s output and project the average of these through the temporal head for video classification. We only train our transformation (to generate temporal prompts) module and temporal class token. The final loss during training is defined on temporal prompts only. Therefore, we mention in Fig. 1, that the refined spatial tokens output by the model are ignored as those represent the image/spatial solution and are not required during the training stage.
>
> **The difference between the term “token” and output of the transformer model**
>
> Thank you for pointing this out. We associate the term class token for the learnable vector (linear layer) [1] and represented as $I_{cls}$, while the class token obtained at the model’s output is referred to as a refined class token and represented as $\tilde{I}_{cls}$. Similarly, we declare notation for video tokens as well (Section 3.1, Preliminaries). We use these notations to define loss functions used during attacks in Section 3.2.
>
> **Is self-supervised L_ss - that unsupervised objective?**
>
> Yes, we present a definition of the self-supervised loss $\mathcal{L}_{ss}$ in Section 3.2 and changed the “unsupervised” to “self-supervised” for consistency.

---

> > ### Author Response · Authors · 2022-11-18
> > **Response to Reviewer yaNd (2/3)**
> >
> > **Experimental observations and conclusions.**
> >
> > **Analysing CLIP:** Our results indicate that adversarial perturbations found via the CLIP vision encoder show lower transferability as compared to similar models (in terms of network design) such as DINO [2] and Deit-B [1]. We further inspect this behavior in Appendix D (**please see Table 9 and Fig. 5**). We observe that CLIP vision features are aligned towards textual features and as a result adversaries found within CLIP models are textually biased. We observe the textual patterns emerging within adversarial perturbations (**Fig. 5 in Appendix D**). These perturbations are optimal to fool CLIP but not that meaningful for vision models trained without text supervision. We quantify this as well by measuring cosine similarly between visual features of CLIP vision encoder, DINO, and Deit-B with CLIP textual features produced for text labels (**Table 9 in Appendix D**). These observations further explain the lower performance of the ensemble of different training frameworks as compared to an ensemble of different networks in Fig. 4 in our paper.
> >
> > The following table presents an analysis of textual bias within vision encoders studied in our work.
> > | | | |
> > | -- | :-: | :-:  |
> > | Vision Encoder | Text Encoder | Cosine Similarity |
> > | CLIP     | CLIP      | 0.31     |
> > | Deit-B   | CLIP        | 0.005     |
> > | DINO    | CLIP        | 0.002     |
> >
> > _Textual Bias of Vision Encoders: Results are reported on 1000 randomly selected images (one per class) from ImageNet validation set. We measure the similarity between the refined class token output by vision encoders with textual features output by CLIP text encoder for text label input as ``a photo of a \{class name\}"._
> >
> > **Self-supervised Inductive Biases boost Adversarial Transferability:** We observe that our temporal prompts focus on action across different frames for a video as compared to the spatial class token with in pre-trained image models (**Fig. 6 in Appendix E**). The temporal prompts with in self-supervised image model DINO with object localization ability (via self-attention) show higher transferability as compared to supervised (Deit) or text supervised (CLIP) models (Tables 1 and 2 in our paper).
> >
> > **Dynamic cues within Image Models:**  As recommended by reviewer TDqC, we provide inference results on shuffled video frames to establish robustness to temporal variations in Appendix F (**Table 10**). We further show that our method improves by increasing the number of temporal tokens during training in Appendix G (**Table 11**) (experiment suggested by reviewer TDqC).
> >
> > The following table presents results on the robustness of our models against temporal variations.
> > | | | |  | | |
> > | --| :-: | :-:  |:-:  |:-:  |:-:  |
> > | Model | Temporal Prompts | UCF | HMDB  | K400 | SSv2|
> > | Deit-B     | ☑       | 80.31$\pm$ 0.33    |  46.2$\pm$0.22 | 57.1$\pm$0.35| 16.9$\pm$0.52|
> > | DINO    | ☑         |    78.40$\pm$0.35  |46.1$\pm$0.19 |55.7$\pm$0.21|17.1$\pm$0.32|
> > | CLIP    | ☑         |   85.6 $\pm$ 0.23  | 55.0 $\pm$ 0.18 | 66.2 $\pm$ 0.33| 19.1 $\pm$ 0.21|
> >
> > _Robustness to Temporal Variations: We repeat inference results three times on a single view with 8  randomly sampled and shuffled frames. Our adapted image models show higher robustness to such temporal variations._

---

> > > ### Author Response · Authors · 2022-11-18
> > > **Response to Reviewer yaNd (3/3)**
> > >
> > >
> > > The following table shows the effect of the number of temporal prompts on adversarial transferability.
> > > |  | | |  | |  |  | | |  |
> > > | -- | :-: | :-: | :-: | :-: | :-: | :-: | :-: | :-: | :-: |
> > > |        |                  | TimesFormer | TimesFormer | TimesFormer | TimesFormer | ResNet3D | ResNet3D | ResNet3D | ResNet3D |
> > > | Model  | Temporal Prompts | UCF         | HMDB        | K400        | SSv2        | UCF      | HMDB     | K400     | SSv2     |
> > > | Deit-B |       8           |       55.5      |      12.6       |      40.6        |    16.8        |     37.3     |      11.4    |     21.5     |    13.3      |
> > > |        |             16     |     51.4        |      10.6       |        38.7     |       15.3      |     35.2     |     9.9     |       20.6   |      15.5    |
> > > |        |             32     |      **21.5**       |       **6.5**       |         **23.1**     |      **10.4**       |      **19.6**     |   **6.0**        |       **13.9**    |      **11.2**     |
> > > |        |                  |             |             |            |             |          |          |          |          |
> > > | DINO   |         8         |      45.2       |    7.3          |       35.8      |      15.8       |     32.2     |    10.6      |     18.3     |  12.0        |
> > > |        |            16      |       42.9      |      6.0     |        35.3       |         14.5    |       29.7   |       8.8   |        18.1  |    11.5      |
> > > |        |              32    |       **30.2**       |        **5.7**    |        **22.6**       |          **8.3**    |      **15.3**     |       **4.9**    |        **10.0**   |       **7.3**    |
> > >
> > > _Effect of Number of Temporal Prompts on Adversarial Transferability: We observe non-trivial gains in adversarial transferability as we increase our temporal prompts. This indicates that the more the temporal information the better the attack transferability (DIM) of our approach (lower is better)._
> > >
> > > **Model Size and Adversarial Transferability:** Finally, we observe that increasing the surrogate model size can increase the adversarial transferability (Tables 1 and 2 in our paper).
> > >
> > > We include these observations in the relevant sections and appendices of the paper (highlighted in red).
> > >
> > > **Reproducibility**
> > >
> > > Please note, we provide all the necessary details to reproduce our work in the reproducibility statement. Further, a well-documented code along with our adapted (trained) models for videos will be made publicly available. We will also release the dataset information used during the attacks as well.
> > >
> > > **Prior work**
> > >
> > > To the best of our knowledge and as pointed out by the reviewer QXGV, this is the very first work to use an image model with dynamic cues for adversarial transferability for videos. Please note that our approach based on temporal cues is complementary to the existing attacks as we demonstrated their effectiveness for different attacks in our paper.
> > >
> > > **References**
> > >
> > > [1]  Hugo Touvron et. al “Training data-efficient image transformers & distillation through attention”, ICML, 2021
> > >
> > > [2] Mathilde Caron et. al "Emerging properties in self-supervised vision transformer”, ICCV, 2021
> > >
> > > [3] Alec Radford et.al “Learning transferable visual models from natural language supervision”, ICML, 2021
> > >
> > > [4] Ashish Vaswani et. al  “Attention Is All You Need”, NeurIPS, 2017.
> > >
> > > [5] Naseer et. al “Cross-Domain Transferability of Adversarial Perturbations”, NeurIPS, 2019

---

### Author Response · Authors · 2022-11-18
**Thank you for the valuable comments**

We thank all the reviewers (yaNd, iNv7, TDqC, QXGV) for the positive feedback and appreciate the detailed comments to improve our work. **Reviewer-yaNd:** “The problem is well-motivated. Results are promising. Evaluation is thorough  across  datasets, surrogate models, adversarial attacks, resolutions, and ensembling techniques”. **Reviewer-iNv7:** “The problem studied in this paper is interesting and important in practice. The proposed method is intuitive, simple and seems effective. The idea of mimicking dynamic cues on static images with different spatial scales is interesting”. **Reviewer-TDqC:** “Incorporating temporal prompts appears to be a promising strategy as it significantly improves the transferability of attacks. The proposed method has a small computational overhead”. **Reviewer-QXGV:** “The first work that tries to transfer adversarial examples from image models to video models using prompt tuning. Approach empirically works well”.

**Our codes and trained models will be publicly released.** We summarize the salient features of the proposed approach below:

* Our work highlights the adversarial transferability from image to video or multi-view models.

* We introduce dynamic cues within frozen image models without losing the original image representation (e.g. generalization on ImageNet). Both image and video representations enhance adversarial transferability from our adapted image models.

* Our approach simply augments the existing adversarial attacks developed for image models.

* We analyze three types of training schemes (supervised, self-supervised, and text-supervised) and highlight new insights into the adversarial space of vision-language models.

* Our approach exhibits state-of-the-art adversarial transferability on black-box image and video models.

---

### Author Response · Authors · 2022-12-07
**We hope the reviewers' concerns are addressed.**

We hope the reviewers' concerns are addressed satisfactorily. If there are any further questions or comments, we look forward to hearing them and will be happy to respond.

---

### Decision · Program_Chairs · 2023-01-20

**Decision:**

Accept: poster

**Justification For Why Not Higher Score:**

All the reviewers are lukewarm on the paper.

**Justification For Why Not Lower Score:**

None of the reviewers bring up substantial concerns about the relevance, novelty, or contribution of the work. Reviewers are lukewarm due to perceived issues with the presentation. When I looked at the paper, I found the writing could certainly be improved but I didn't find it a blocker in understanding the content.

**Metareview: Summary, Strengths And Weaknesses:**

The paper studies transferability of adversarial examples from white-box image recognition models to black-box video recognition models. It alters a pre-trained image recognition model to produce “temporal tokens” that are concatenated with the spatial tokens in a ViT. The resulting model is used to produce adversarial examples that attain higher attack success rate against the aforementioned black-box models.

The reviewers all recognize the relevance of the learning setting studied, the novelty of the proposed approach, and the efficacy of the proposed method. The reviewers also expressed concerns about the clarity of presentation and missing comparisons with related work. The latter concerns were largely addressed by the additional experimental results provided as part of the discussion.

Overall, the AC finds that, while the presentation of the work can be improved, the paper is of interest to the ICLR audience and recommends it for acceptance. The authors are encouraged to improve the presentation in the camera-ready version of the manuscript.

**Note From Pc:**

if the above contains the word "oral" or "spotlight" please see: "oral" presentation means -> notable-top-5% and "spotlight" means -> notable-top-25%. As stated in our emails, we are disassociating presentation type from AC recommendations

**Summary Of Ac-Reviewer Meeting:**

N/A